

# PyTrx: A Python toolbox for deriving velocities, surface areas and line measurements from oblique imagery in glacial environments

Penelope How[1,2], Nicholas R. J. Hulton[1,2], and Lynne Buie[1]

[1]Institute of Geography, School of GeoSciences, University of Edinburgh, Edinburgh, UK
[2]Department of Arctic Geology, University Centre in Svalbard, Longyearbyen, Norway

**Correspondence:** Penelope How (p.how@ed.ac.uk)

**Abstract.** Terrestrial time-lapse photogrammetry is a rapidly growing method for deriving measurements from glacial environments because it provides high spatio-temporal resolution records of change. However, glacial photogrammetry toolboxes are limited currently. Without prior knowledge in photogrammetry and computer coding, they are used primarily to calculate ice flow velocities or to serve as qualitative records. PyTrx (available at https://github.com/PennyHow/PyTrx) is presented here as

a Python-alternative toolbox to widen the range of photogrammetry toolboxes on offer to the glaciology community. The toolbox holds core photogrammetric functions for point seeding, feature-tracking, image registration, and georectification (using a planar projective transformation model). In addition, PyTrx facilitates areal and line measurements, which can be detected from imagery using either an automated or manual approach. Examples of PyTrx's applications are demonstrated using time-lapse imagery from Kronebreen and Tunabreen, two tidewater glaciers in Svalbard. Products from these applications include ice flow

velocities, surface areas of supraglacial lakes and meltwater plumes, and glacier terminus profiles.

## 1 Introduction

Terrestrial time-lapse photogrammetry has proved to be a viable approach for obtaining high spatio-temporal resolution observational records from tidewater glaciers (e.g., Ahn and Box, 2010; Rosenau et al., 2013; James et al., 2014; Pętlicki et al., 2015). It provides adequate spatial resolution, and the temporal frequency of data-capture is flexible and relatively easy to control.

However, time-lapse photogrammetry remains an under-used technique in glaciology because there are few publicly-available toolboxes for deriving real-world, meaningful measurements from terrestrial imagery. The majority of these toolboxes have been programmed in one computing language and, without prior knowledge in photogrammetry and computer coding, their applications are largely limited to calculating glacier surface velocities (e.g., Kääb and Vollmer, 2000; Messerli and Grinsted, 2015; James et al., 2016). More toolboxes are therefore needed to further this technique and its glaciological applications, and

add to the data products that can be obtained from time-lapse imagery.

     PyTrx (short for 'Python Tracking') is a new toolbox, which is presented here to widen the range of photogrammetric toolboxes on offer to the glaciology community, and also expand the types of measurements that can be derived from time-lapse imagery (available at https://github.com/PennyHow/PyTrx). PyTrx has been developed with glaciological applications in mind, with functions for deriving surface velocities via a sparse feature-tracking approach, surface areas (e.g. supraglacial



lakes and meltwater plume expressions) with automated area detection, and line profiles (e.g. glacier terminus position) with a manual point selection method.

## 2 Background

Photogrammetry is defined broadly as the extraction of measurements from photographs, which may be captured either from
above (i.e. aerial) or from the ground (i.e. terrestrial). In recent years, photogrammetry has moved away from traditional techniques (e.g., Finsterwalder, 1954; Kick, 1966), with the introduction of digital cameras and computers with high processing powers.

Terrestrial photogrammetry is a rapidly growing technique in glaciology as a result of its expanding capabilities, with applications in monitoring glacier surface conditions (Parajka et al., 2012; Huss et al., 2013), supraglacial lakes (Danielson and
Sharp, 2013), meltwater plume activity (Schild et al., 2016; How et al., 2017; Slater et al., 2017), and calving dynamics (Kaufmann and Ladstädter, 2008; Ahn and Box, 2010; James et al., 2014; Whitehead et al., 2016; Pętlicki et al., 2015; Medrzycka et al., 2016; Mallalieu et al., 2017; How et al., In Review). A prevailing application has been in deriving glacier surface velocity from sequential terrestrial imagery using a technique called feature-tracking; as it offers highly detailed (both spatially and temporally) records (e.g., Fox et al., 1997; Maas et al., 2006; Dietrich et al., 2007; Eiken and Sund, 2012; Heid and Kääb, 2012;
Rosenau et al., 2013). A handful of software has been developed to perform feature-tracking through terrestrial time-lapse imagery (e.g., Kääb and Vollmer, 2000; Messerli and Grinsted, 2015; James et al., 2016).

A key problem is the increasing demand for efficient photogrammetry software, which can execute large-batch processing quickly. The future of its application in glaciology lies in its valuable ability to examine different aspects of the glacier system simultaneously, such as glacier velocity, fjord dynamics, surface lake drainage and calving dynamics. These can be studied
using different image capture frequencies and over different lengths of time. To achieve this, the glaciology community needs a greater range of robust photogrammetry methods which have been developed specifically for applications in glaciology.

Here, a new time-lapse photogrammetry toolbox is presented with specific applications in glaciology. PyTrx has been developed to further terrestrial time-lapse photogrammetry techniques in glaciology and address the issues outlined previously. The software is coded in Python, an open-source computing language, which is freely available and easily accessible to be-
ginners in programming. PyTrx is capable of producing velocities, surface areas and distances from monoscopic time-lapse set-ups. The common photogrammetry methods used in glaciology will be outlined subsequently, followed by PyTrx's key features and differences. PyTrx's capabilities will be demonstrated and evaluated using time-lapse imagery from Kronebreen and Tunabreen, two tidewater glaciers in Svalbard.

## 3 Common photogrammetric methods

Current photogrammetry software can generally be divided into those that perform feature-tracking algorithms such as IM-CORR (Scambos et al., 1992), COSI-Corr (Leprince et al., 2007), and CIAS (Kääb and Vollmer, 2000; Heid and Kääb, 2012);



and those that perform image translation functions such as Photogeoref (Corripio, 2004), PRACTISE (Härer et al., 2016), Agisoft PhotoScan, and PhotoModeler. A common limitation is that few pieces of software unite all the photogrammetry processes needed to compute real world measurements from terrestrial time-lapse imagery (i.e. distance, area, velocity, and volume). There are a handful of toolboxes that provide functions for all of these processes, such as the Computer Vision

System toolbox for Matlab and the OpenCV toolbox for C++ and Python. However, these are merely given as stand-alone algorithms and a significant amount of time and knowledge is needed to produce the desired measurements and information.

ImGRAFT (available at https://imgraft.glaciology.net) and Pointcatcher (available at https://www.lancaster.ac.uk/...pointcatcher.htm) were the first toolboxes that were made publicly available and contain all the processes needed to obtain velocities from terrestrial time-lapse imagery in glacial environments (Messerli and Grinsted, 2015; James et al., 2016). Both are Matlab-based

toolboxes using algorithms from the Computer Vision System toolbox, developed specifically for glaciological applications. These software follow a similar workflow and the steps involved will be discussed subsequently.

### 3.1    Image processing

Images need to display consistent conditions throughout an image series to gain the best photogrammetric measurements. Therefore, images are commonly enhanced in order to achieve this. Images can be enhanced individually, but this is often

time-consuming when handling large image sets. Batch processing is more commonly utilised to enhance images in a time-lapse sequence.

The most straightforward types of enhancements are point operators, where the output value of each pixel in an image depends solely on the corresponding input pixel value, and a given parameter in some cases (Szeliski, 2010). These include brightness and contrast adjustments, colour corrections. These adjustments do not directly improve the quality of an image

though, as changes affect pixel hue and saturation as well as apparent intensity.

Histogram equalisation (also called global histogram equalisation) is generally used to achieve suitable pixel values in an automated manner. An intensity mapping function is calculated by computing the cumulative distribution function ($c(I)$) with an integrated distribution ($h(I)$) and the known number of pixels in the image ($N$) (Solem, 2012):

$$c(I) = \frac{1}{N} \sum_{i=0}^{I} h(I) = c(I-1) + \frac{1}{N} h(I) \tag{1}$$

This reduces the range of pixel values in an image, and smooths drastic changes in lighting and colour.

### 3.2    Displacement analysis

Displacements are measured through a sequence of images using a technique called feature-tracking, by which pixel intensity features are matched from one image to another (Szeliski, 2010; Solem, 2012). Pixel-intensity features are defined in the image plane either as points or pixel regions (also referred to as templates), producing spot measurements and continuous surface

measurements respectively. These two approaches are also referred to as sparse feature-tracking and dense feature-tracking.



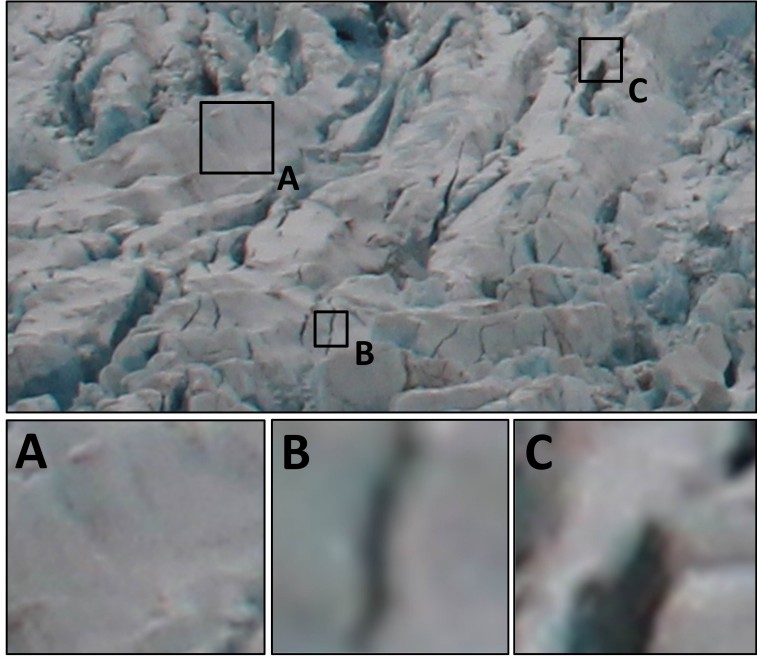

**Figure 1.** Point seeding coherency demonstrated using regions of a time-lapse image of a crevasse field. The crevasse field (top image) is a subset of a time-lapse image from Kronebreen, Svalbard. The regions highlighted from this subset are examples of a homogenous area (A), an edge feature (B), and a corner feature (C). A has a very indistinctive pixel pattern so would be unable to track from image to image. B has a distinct boundary between different pixel intensities, but tracking may drift along the boundary over time. C is a very distinctive pixel pattern with a defined point that is easy to track.

Feature-tracking can be conducted by creating and tracking exclusively between each image pair in an image set (e.g., Messerli and Grinsted, 2015), or by tracking continuously through the image set using the same pixel-intensity features (e.g., James et al., 2016). Whilst tracking continuously enables the calculation of cumulative feature displacements, tracking between each image pair reduces the risk of error propagation from false matches.

5     Corner objects provide good features to track for sparse feature-tracking methods because they provide distinctive pixel-intensity distributions that can be matched in the subsequent image. This is demonstrated in Fig. 1, where corner features (e.g. a crevasse corner or debris corner) within an image of a crevasse field prove more distinctive than a homogeneous surface (e.g. bare ice) or an edge feature (e.g. a crevasse edge). These features are marked as points on an image, which can be matched and/or tracked in subsequent images. The creation of these points is called point seeding. Point seeding can be conducted on a
10   manual basis, but this is often time-consuming when handling large image sets (e.g., How, 2013). Automated corner detection methods utilise the high contrast in pixel values to identify the maximum variation in a given region of the image (Szeliski, 2010). There are numerous corner detection methods available for this process, such as the Harris corner detection method (Harris and Stephens, 1988) and the FAST (Features from Accelerated Segment Test) corner detection method (Rosten and

Drummond, 2006). The main difference in these methods is precision and efficiency, with trade-offs between the two (Solem, 2012).

When tracking between an image pair, the two images have been referred to in many ways, such as 'image A' and 'image B' (e.g., Messerli and Grinsted, 2015), and the 'reference' and 'destination'/'search' images (e.g., Ahn and Box, 2010; James et al., 2016). The terms reference image and destination image will be used subsequently; the reference image being where points are seeded or templates are defined, and the destination image being where a point/template is matched to. Normalised cross-correlation is a common approach for automated feature-tracking, with cross-correlation referring to the correlation between two signals (i.e. the pixel intensity distribution in two images). This technique is applied in both sparse and dense feature tracking in a similar manner, using the pixel intensity distribution in a window around a given point or a template in the reference image ($T$) (Szeliski, 2010; Solem, 2012):

$$R(x,y) = \frac{\sum_{x',y'}\left(T(x',y')\cdot I(x+x',y+y')\right)}{\sqrt{\sum_{x',y'}T(x',y')^2\cdot\sum_{x',y'}I(x+x',y+y')^2}} \tag{2}$$

Where $R$ is the correlation between the reference template and the destination image, and I is the destination image. The function is applied to each possible position in the image $(x,y)$, thus defining the correlation at every point $(x',y')$. The highest correlation is defined as the best match between the reference template and the destination image. The correlation at each point in the image can also be determined using different correlation methods such as the normalised square difference, the least square sum and the least difference methods (Szeliski, 2010).

Tracking coherency through long-duration sequences is frequently subject to severe lighting discrepancies and shadowing which cause false motion. In such applications there is a large significance on image selection. Images with similar lighting and limited shadowing variation must be selected, which may limit the temporal resolution of the collected data. Glacial features which provide good tracking points/templates, such as debris features, can evolve over time which introduce additional displacements (e.g., How, 2013). This can be especially limiting for area-based tracking (e.g., Messerli and Grinsted, 2015). In sparse point tracking, there is a heavy reliance on the number and distribution of points which reduces the replicability of results (Fox et al., 1997; James et al., 2016). It is the factors outlined here that need to be considered when choosing a feature-tracking approach.

### 3.3 Motion correction

During image acquisition, the time-lapse camera platform is often subject to movement caused by instabilities in the installation, wind, ground heave, thermal expansion of the tripod, and animal/human intervention. This introduces false motion to the measurements derived from an image sequence, which need to be corrected for in order to make accurate measurements through sequential imagery. This process is referred to as image registration.

Feature-based registration methods are more commonly used for glacial environments due to large variations in lighting and glacier surface evolution over time, especially over long-duration sequences (e.g., Messerli and Grinsted, 2015; James et al., 2016). Feature-based registration aligns the images by tracking static feature points. These can be natural features, such as





mountain peaks (e.g., James et al., 2016), or man-made targets (e.g., Dietrich et al., 2007). Observed movement of these points signify false motion.

Between 20 and 30 coherent natural static feature points can represent false motion effectively. Ideally, these would be distributed evenly across the image plane. Control points in the foreground of an image are more sensitive and can be better

for constraining camera rotation angles (Eiken and Sund, 2012), but equally heavy reliance on these can introduce excessive noise and random pixel variation (James et al., 2016).

The two-dimensional pixel displacements between point pairs are subsequently used to calculate motion in the camera, which is represented in three-dimensional space. This translation to three-dimensional space is achieved using a transformation model. There are a number of transformation models in existence, such as isometries, similarity transformations, and affine

transformations. Planar projective transformations are used typically for time-lapse photogrammetry in glaciology because they utilise homogeneous coordinates and therefore translate an original planar to a continuous surface. This is also a technique typically used in georectification (see subsequent section for more details).

The point pairs from the static point features are used to map the destination image to the reference image. This map is also referred to as the homography $(H)$, and is computed as follows:

$$\begin{bmatrix} x' \\ y' \\ w' \end{bmatrix} = \begin{bmatrix} h_1 & h_2 & h_3 \\ h_4 & h_5 & h_6 \\ h_7 & h_8 & h_9 \end{bmatrix} \begin{bmatrix} x \\ y \\ w \end{bmatrix} \quad or \quad x' = Hx \tag{3}$$

Where the $h$ values correspond with the homogeneous transformation of each point within the planar, which is used to translate coordinates from the original image $(x, y, w)$ to the destination image $(x', y', w')$. In other words, coordinates in the destination $(x')$ are represented by the homography and the corresponding coordinates in the reference image $(Hx)$ (Hartley and Zisserman, 2004).

The homography encapsulates the three-dimensional rotation of the camera platform as movement around its horizontal (yaw), vertical (pitch) and optic axes (roll) (also referred to as omega, phi, and kappa by James et al., 2016). Rotation that cannot be accounted for from the two-dimensional displacements is represented as a root-mean-square (RMS) residual pixel value, which is used as a measure of uncertainty. The output rotations can be applied to correct false motion from feature track measurements, which will improve the signal-to-noise ratio.

## 3.4 3D conversion

Image translation is the process by which measurements in the image plane are translated to real-world measurements. There are several approaches to image translation, the two main ones in glaciology being scale factoring and georectification. A scale factor describes the absolute distance per pixel in an image at a given distance. However, this assumes that the measured displacements are precisely perpendicular to the direction of the camera (e.g., Ahn and Box, 2010). With georectification, the

image plane is mapped directly to a real-world coordinate system, and this is more commonly used for measuring displacements that are at an angle to the camera.




The planar projective transformation technique described previously (Equation 3) is also used in image georectification. A homography model is calculated that represents the translation from the image plane to the three-dimensional environment (Hartley and Zisserman, 2004). This is determined using an assortment of information about the three-dimensional environment and how the camera captures this. A Digital Elevation Model (DEM) is typically used to represent the three-dimensional

environment. Ground Control Points (GCPs) are point locations in the image plane with corresponding real-world coordinates, which are used as pinning points to map the image to a known coordinate system.

Geometric camera calibration is used to model how the camera captures the three-dimensional environment. There are numerous models which define this translation (Solem, 2012). The precedent model used in glaciology utilises the extrinsic $(R, t)$ and intrinsic $(K)$ information about the camera:

$$
\quad P = \begin{bmatrix} R \\ t \end{bmatrix} K \tag{4}
$$

Where $P$ is the camera matrix that mathematically represents the translation between the three-dimensional world scene and the two-dimensional image; $R, t$ are the extrinsic camera parameters that represent the location of the camera in three-dimensional space; and $K$ are the intrinsic camera parameters that represent the conversion from three-dimensional space to a two-dimensional image plane.

The extrinsic and intrinsic camera parameters can be considered as separate matrices (Solem, 2012). The extrinsic camera matrix consists of a $3 \times 3$ matrix that represents camera rotation $(r)$ and a column vector that represents camera translation $(t)$:

$$
\begin{bmatrix} R \mid t \end{bmatrix} = \left[ \begin{array}{ccc|c} r_{1,1} & r_{1,2} & r_{1,3} & t_1 \\ r_{2,1} & r_{2,2} & r_{2,3} & t_2 \\ r_{3,1} & r_{3,2} & r_{3,3} & t_3 \end{array} \right] \tag{5}
$$

The intrinsic camera matrix $(K)$ is a $3 \times 3$ matrix that contains information about the focal length of the camera in pixels $(f_x, f_y)$, the principal point in the image $(c_x, c_y)$ and the camera skew $(s)$ (Solem, 2012):

$$
\quad K = \begin{bmatrix} f_x & s & c_x \\ 0 & f_y & c_y \\ 0 & 0 & 1 \end{bmatrix} \tag{6}
$$

The given focal length for a fixed focal length lens can be assumed to be precise because there is a limited chance of lens drift. It is advised to calculate the focal length for images captured with zoom lenses and compact cameras for greater accuracy. The principal point is also referred to as the optical centre of an image, and describes the intersection of the optical axis and the image plane. Its position is not always the physical centre of the image due to imperfections produced in the camera

manufacturing process (Busch et al., 2014); this difference is known as the principal point offset (Hartley and Zisserman, 2004). The skew coefficient is the measure of the angle between the $xy$ pixel axes, and is a non-zero value if the image axes are not perpendicular (i.e. a 'skewed' pixel grid).




The intrinsic camera matrix $(K)$ assumes that the system is a pinhole camera model and does not use a lens to gather and focus light to the camera sensor (Solem, 2012). Camera systems that include a lens introduce distortions to the image plane. These distortions are a deviation from a rectilinear projection, in which straight lines in the real world remain straight in an image. These distortions ineffectively represent the target object in the real world, and therefore distortion coefficients

$(k_1, k_2, p_1, p_2, k_3 \ldots k_8)$ are needed to correct for this. These coefficient values correct for radial $(k_1, k_2, k_3 \ldots k_8)$ and tangential $(p_1, p_2)$ distortions (Hartley and Zisserman, 2004):

$$
x_{corrected} = x'\left(1 + k_1 r^2 + k_2 r^4 + k_3 r^6 \ldots + k_8 r^{16}\right)
$$
$$
y_{corrected} = y'\left(1 + k_1 r^2 + k_2 r^4 + k_3 r^6 \ldots + k_8 r^{16}\right)
$$

$$(7)$$

$$
x_{corrected} = x' + \left[2p_1 xy + p_2\left(r^2 + 2x^2\right)\right]
$$
$$
y_{corrected} = y' + \left[p_1\left(r^2 + 2y^2\right) + 2p_2 xy\right]
$$

$$(8)$$

Where $x', y'$ are the uncorrected pixel locations in an image, and $x_{corrected}, y_{corrected}$ are their corrected counterparts. Radial distortion arises from the symmetry of the camera lens whilst tangential distortion is caused by misalignment of the camera lens and the camera sensor. Radial distortion is the more apparent type of distortion in images, especially in wide angle images, and those containing straight lines (e.g. skyscraper landscapes) which appear curved. Severe tangential distortion can visibly alter the depth perception in images.

The camera matrix and these distortion coefficients can be computed using a set of calibration images taken with the camera that is being used for photogrammetry purposes (Hartley and Zisserman, 2004). These calibration images must be of an object with a known geometry or contain known coordinates (Heikkila and Silven, 1997; Zhang, 2000). A commonly used object is a black and white chessboard, with the positioning and distance between the corners used as the $x', y'$ coordinates (Solem, 2012). Other objects which can be used are grids of symmetrical circles (with the centre of each circle forming the $x', y'$ coordinates),

and targets which are specified by programs that perform camera calibration (e.g. Agisoft Photomodeller).

The location of the camera and its initial pose (yaw, pitch, roll) are needed to accurately position the camera within the three-dimensional environment. Yaw, pitch and roll are difficult to accurately measure in the field and certain photogrammetry toolboxes calculate and refine yaw, pitch and roll automatically, such as ImGRAFT (Messerli and Grinsted, 2015). Camera pose can also be calculated using the principal point (represented as a GCP), along with additional corresponding GCPs to

measure the three axes (Fig. 2). The y-axis position of the principal point is used to calculate yaw as an azimuth bearing (Fig. 2A). The $xyz$ position of the principal point provides an elevation comparison to the camera location, which defines the pitch rotation (Fig. 2B). A GCP along the same x-axis position as the principal point is used to calculate roll (Fig. 2C), signified by the apparent change in elevation (Addison, 2015).





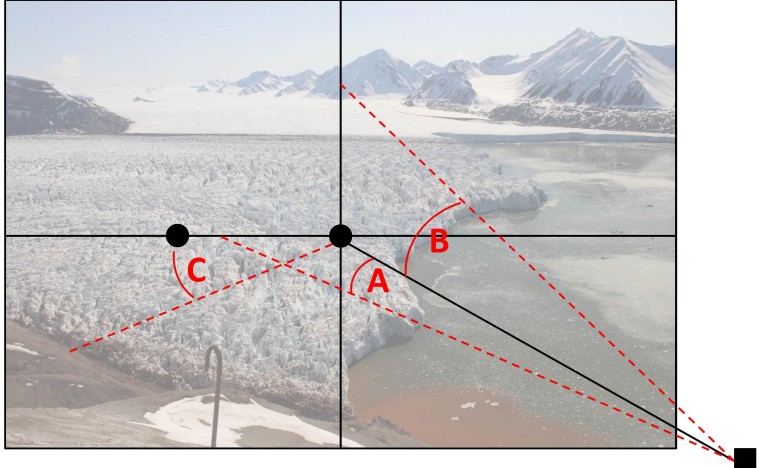

**Figure 2.** Diagram demonstrating yaw (A), pitch (B), and roll (C) from an image, knowing the camera location (denoted by the square marker) and two GCPs along the x-axis of the principal point (denoted by the two circle markers). The underlying time-lapse image is of the terminus of Kronebreen, Svalbard. Figure adapted from Addison (2015).

## 4    Features and Applications of PyTrx

PyTrx (available at https://github.com/PennyHow/PyTrx) has been developed to further terrestrial time-lapse photogrammetry techniques in glaciology, offer an alternative to the toolboxes currently available, and address the issues outlined previously. PyTrx is aimed at beginners in programming, with the files associated with the toolbox written with object-oriented design (i.e.

with classes that contain the associated methods and functions). The toolbox follows a similar workflow to that outlined in the previous section.

### 4.1    Field set-up

Examples are given throughout this section, which demonstrate the capabilities of PyTrx and its applications in glaciology. These examples use time-lapse imagery collected from Kronebreen (78.8°N, 12.7°E, Fig. 3B) and Tunabreen (78.3°N, 12.3°E,

Fig. 3C), which are two tidewater glaciers in Svalbard (Fig. 3A). These time-lapse camera systems consisted of a Canon 600D/700D camera body and a Harbortronics Digisnap 2700 intervalometer, which were powered by a 12 V DC battery and a 10 W solar panel. An assortment of camera lenses was used to allow for flexibility in coverage. These were primarily made up of EF 20 mm f/2.8 USM and EF 50 mm f/1.8 II prime lenses.

The camera parts and the battery were encased in waterproof Peli Case boxes. The camera enclosures were modified to have

a porthole that could hold a sheet of optical glass between two steel frames, through which the camera could take photographs. The camera boxes were fixed on tripods, which were anchored by digging the tripod legs into the ground, burying the tripod legs with stones, and/or drilling guide wires into the surrounding bedrock. An example of one of these set-ups is shown in Fig. 3D.



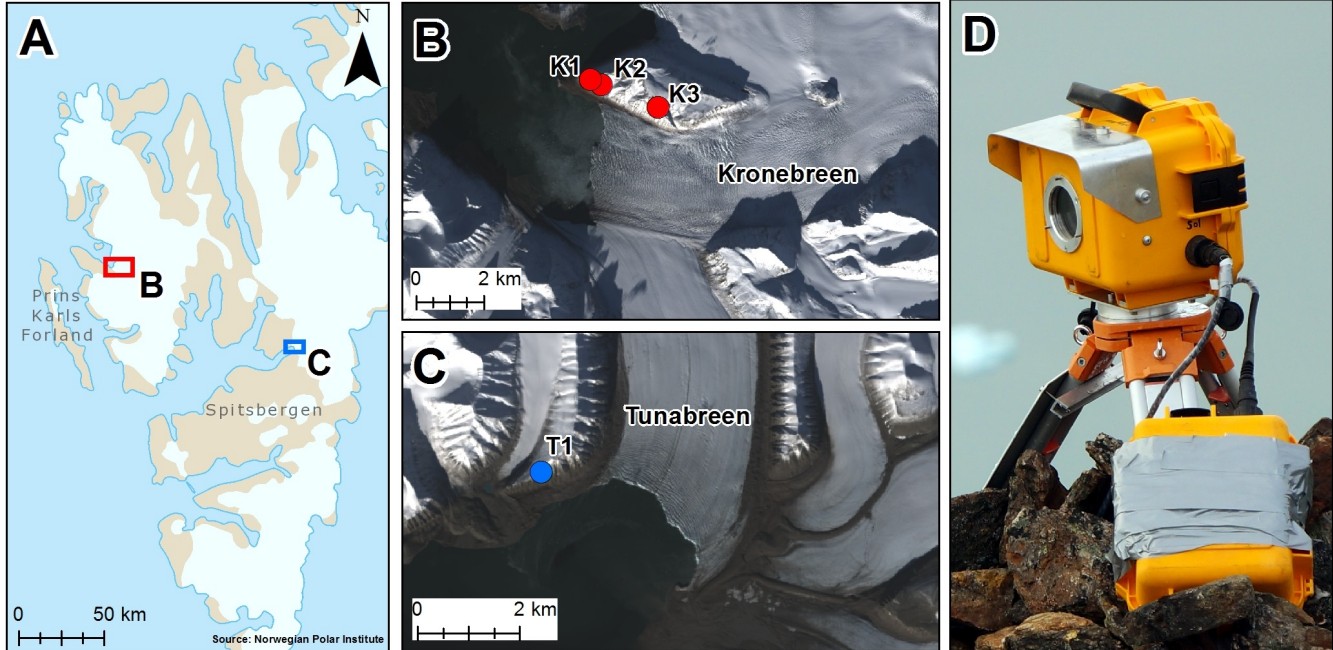

**Figure 3.** Maps showing the Svalbard archipelago (A); Kronebreen (B) and Tunabreen (C) with numbered camera sites installed over the 2014 and 2015 melt seasons; and an example of one of the time-lapse camera installations in the field (D).

Accurate locations for each of the time-lapse cameras were measured using a Trimble GeoXR GPS rover to a SPS855 base station, which was positioned ~15 km away. Positions were differentially post-processed in a kinematic mode using the Trimble Business Centre software, given an average horizontal positional accuracy of 1.15 m and an average vertical positional accuracy of 1.92 m.

GCPs were determined for each camera set-up from known xyz locations that were visible in the field-of-view, and camera pose (yaw, pitch roll) was determined using the approach demonstrated in Fig. 2. Each camera (and lens) was calibrated using the camera calibration functions in the Matlab Computer Vision Systems toolbox to obtain intrinsic camera matrices and lens distortion coefficient values.

The DEM of the Kongsfjorden area orginates from a freely available DEM dataset provided by the Norwegian Polar Institute,

which was obtained from airborne photogrammetric surveying in 2009 (Norwegian Polar Institute, 2014). The DEM of the Tempelfjorden area originates from ArcticDEM, Scene ID WV01-20130714-1020010 (14th July 2013). These DEMs are distributed with PyTrx in a modified form, with each scene clipped to the area of interest, downgraded to 20 m resolution, and smoothed using a linear interpolation method. In cases where measurements are derived at sea level (e.g. meltwater plume extents, terminus profiles, and calving event locations), all low-lying elevations (< 150 m) have been transformed to 0 m a.s.l.

in order to project them to a flat, homogeneous surface.





## 4.2 Structure of PyTrx

PyTrx is compatible with Python 2.7 releases, and largely utilises the OpenCV (Open Source Computer Vision) toolbox (v3.1 and upwards), which is a free library designed to provide computer vision and machine learning tools that are computationally efficient and operational in real-time applications. The library has over 2500 optimised algorithms including those for mono-

scopic photogrammetry and camera calibration (Solem, 2012). A number of other packages are also used, notably GDAL, Glob, Matplotlib, NumPy, OsGeo, and PIL; and these come pre-installed with most Python distributions such as PythonXY and Anaconda.

PyTrx is distributed as a series of files, which requires a driver script to run. The toolbox consists of six Python files, which handle the main classes and functions:

1. CamEnv.py – Handles the objects associated with the camera environment;

2. DEM.py – Handles the DEM object;

3. Images.py – Handles the objects associated with the image sequence and the individual images within that sequence;

4. Measure.py – Handles the objects associated with making measurements from the images (i.e. velocities, areas, and distances);

5. FileHandler.py – Contains all functions called by an object to import and export data;

6. Utilities.py – Contains all functions for plotting and presenting data.

Within these files, six class objects perform the core photogrammetry processes that were outlined in the previous section: *CamImage*, *ImageSequence*, *Velocity*, *Area*, *Length*, and *CamEnv*. They operate according to the workflow presented in Fig. 4.

The key features of PyTrx, which are different from the general techniques discussed previously, will be outlined compre-

hensively in the subsequent sections with reference to PyTrx's object-oriented workflow.

## 4.3 Image enhancement

*CamImage* (found in Images.py) holds all the information about a single image within an image sequence, making single images easy to call within PyTrx. Each image is represented as a NumPy array in the CamImage class object. Image enhancement processes can be executed within the CamImage object by modifying the NumPy array that represents the image. The image

enhancement methods that are available in PyTrx are histogram equalisation (as presented in Section 3.1) and the extraction of information from a single image band or grayscale.

One problem with histogram equalisation is that the resulting histogram is flat (Solem, 2012). Grayscale, equalised images are used commonly in photogrammetric processing in order to reduce processing time (e.g., James et al., 2016). This means that all the RGB information is flattened and each pixel is assigned one single grayscale value. However, this can alter the image

and its uses for extracting measurements from. For instance, corner coherency can be altered, which can present challenges in





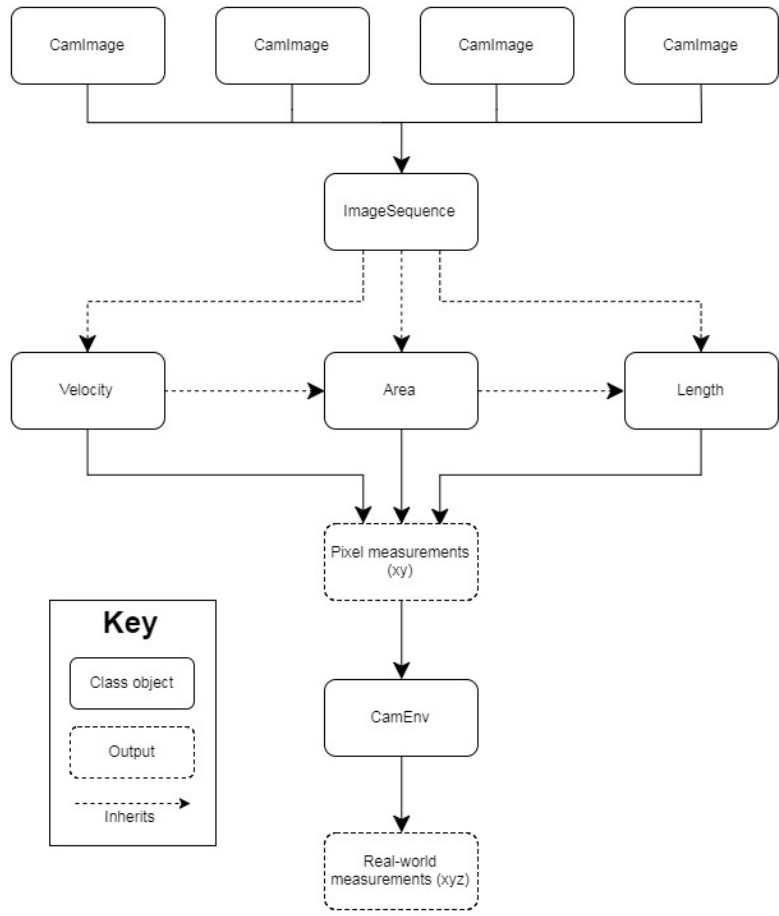

**Figure 4.** PyTrx's workflow, showing how each of the class objects interact with one another.

point seeding, image registration, and feature tracking (Solem, 2012). Additionally, it can make it difficult to distinguish areas of interest based on pixel intensity.

PyTrx has been designed to overcome this limitation by providing a method for extracting information from a specified band of an image. An image can be passed forward either in grayscale, or with one of the RGB bands. Although this is executed in

5 the *CamImage* class object, it can also be defined in the three *Measure* classes (i.e. *Velocity*, *Area*, and *Length*) with the *band* variable. The string inputs *r*, *g*, *b*, and *l* denote whether the red, green, blue, or grayscale bands should be passed forward. This does not affect the processing time drastically, and enables effective detection of areas of interest in images, such as meltwater plumes and supraglacial lakes.

The user can select an image band to suit their applications, and obtain accurate measurements from an appropriate image

10 band. The example in Fig. 5 demonstrates how each selection affects the pixel intensity range associated with a cluster of supraglacial lakes. These surface areas have been detected automatically based on pixel intensity.

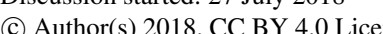

**Figure 5.** An example of PyTrx's ability to the extract pixel information from a specified image band using an example image from Krone-breen camera K3. The image shows a cluster of supraglacial lakes, which were monitored through the 2015 melt season. A shows the original time-lapse image. The yellow box denotes the subset from which pixel information is extracted from and displayed in the subsequent images: grayscale (B), the red image band (C), the green image band (D), and the blue image band (E). The white plotted lines in these subsets show attempts to automatically detect the lake extent. The red image band yields the best detection as it closely follows the lake extent.





Ideally, regions of interest in an image are effectively detected when they are represented by the smallest range in pixel intensity (i.e. a homogeneous surface). With the functionality provided in PyTrx, an assessment can be made to decide which image band facilitates effective detection. In the example presented in Fig. 5, the red image band (Fig. 5C) offers the smallest pixel intensity range and thus the lakes are represented as a homogeneous surface. This proves easiest to define on an automated

basis.

The *Area* class object offers an additional enhancement process to improve the ability to automatically detect areas in images. The image enhancement process within the *Area* class object is a function that uses simple arithmetic to manipulate image brightness and contrast. This is performed in a NumPy array, and the array can either represent an image with pixel values from the red, green or blue band, or from the grayscale image; as specified in the *band* variable. This enhancement

method uses three variables to change the intensity and range of the pixel values:

1. *diff*: Changes the intensity range of the image pixels. This has two outcomes. Either it changes dark pixels to become much brighter and bright pixels become slightly brighter, or it changes dark pixels to become much darker and bright pixels become slightly darker

2. *phi*: Modifies the intensity of all pixel values

3. *theta*: Defines the number of 'colours' in the image by grouping pixel intensity regions together i.e. an input of 3 signifies that all the pixels will be grouped into one of three pixel values

The result better distinguishes areas of interest, and makes it easier for the subsequent detection. See Section 4.6 for more information on how areal measurements are derived from images using PyTrx.

### 4.4 Point seeding

Points are seeded in an image using the Shi-Tomasi Corner Detection method (Shi and Tomasi, 1994), as part of the *good-FeaturesToTrack* function in the OpenCV library. This is based on the Harris Corner Detection method, which evaluates the difference in intensity for a displacement of $(u, v)$ in all directions for a given region of an image $(E)$ (Harris and Stephens, 1988). Points are selected based on the largest intensity differences:

$$E(u,v) = \sum_{x,y} w(x,y) \Big[ I(x+u, y+v) - I(x,y) \Big]^2 \tag{9}$$

Where $w$ is the window function (rectangular or Gaussian) defined as a width and height $(x, y)$ and $I$ is pixel intensity. The first part of the bracketed section, $(I(x+u, y+v))$ defines the shifted intensity, and the second part $(I(x,y))$ calculates the intensity at the centre origin. A scoring function $(R)$ is subsequently used to define whether the pixel-intensity signature represents a corner, a flat area or an edge:

$$R = \lambda_1 \lambda_2 - k \Big( \lambda_1 + \lambda_2 \Big)^2 \tag{10}$$

Where $k$ is a tuneable sensitivity parameter, and $\lambda_1$ and $\lambda_2$ are the eigen values in the $x$ and $y$ axes of a given symmetric matrix (Solem, 2012). This forms a descriptor for the matrix, which can be used to evaluate a given region of an image:





1. A corner is present if $\lambda_1$ and $\lambda_2$ are both large positive values (e.g. Fig. 1C);

2. An edge is present if one of the eigenvalues is large and the other is approximately zero (e.g. $\lambda_1 > 0$ and $\lambda_2 \approx 0$) (e.g. Fig. 1B);

3. A homogenous ('flat') region of the image is present if $\lambda \approx \lambda_2 \approx 0$ (e.g. Fig. 1A)

Changes between $\lambda_1$ and $\lambda_2$ can be amplified by modifying $k$, the sensitivity parameter (Harris and Stephens, 1988). The Shi-Tomasi Corner Detection method built upon this with a further scoring function that ranks the best corner features based on the quality level and the minimum Euclidean distance (Shi and Tomasi, 1994):

$$R = min\Big(\lambda_1\lambda_2\Big) \tag{11}$$

The quality level denotes the minimum quality of a corner and is measured as a value between 0 and 1, and the function returns
the remaining strongest corners.

     An image can be called using the *ImageSequence* object within PyTrx (found in Images.py). An *ImageSequence* object holds the information about a series of *CamImage* objects (as shown in Fig. 4). It references the *CamImage* objects sequentially so that they can be called easily in subsequent processing, such as the selection of single images and image pairs. Images are called in this manner to seed points within. PyTrx attempts to seed 50,000 points (with a quality level of 0.1 and a minimum
Euclidean distance of 3 pixels) in an image. This can be adjusted accordingly throuh PyTrx's *featureTrack* function, which is found in the *Velocity* object in Measure.py. These default settings produce a heavily populated sparse point set in the image plane.

## 4.5   Deriving velocities from feature-tracking

*Velocity* (found in Measure.py) contains all the processing steps for *point seeding*, sparse *feature-tracking*, and *image regis-*
*tration* between image pairs. Points are seeded using the Shi-Tomasi Corner Detection method discussed previously. These seeded points are tracked between image pairs in PyTrx using the Lucas Kanade Optical Flow Approximation method (Lucas and Kanade, 1981). These are verified using a back-tracking technique, and the retained points are reprojected subsequently using the *georectification* functions held within the *CamEnv* class object. This approach is adopted also for *image registration*, which uses control points seeded and tracked between image pairs prior to deriving velocities. An example of this feature-
tracking and georectification functionality is shown in Fig. 6, demonstrated by deriving surface velocities from an image pair captured at Kronebreen.

     Optical Flow is the pattern of apparent motion of an object between two images, caused by the movement of the object or the camera. It is a concept readily employed in video processing to distinguish motion and has been used in motion detection applications to predict the trajectory and velocity of objects (e.g., Baker et al., 2011; Vogel et al., 2012).
Optical Flow is represented as a two-dimensional vector field working on the assumptions that the pixel-intensity distribution of an object does not change between the image pair and the neighbouring pixels display similar motion (Tomasi and Kanade,



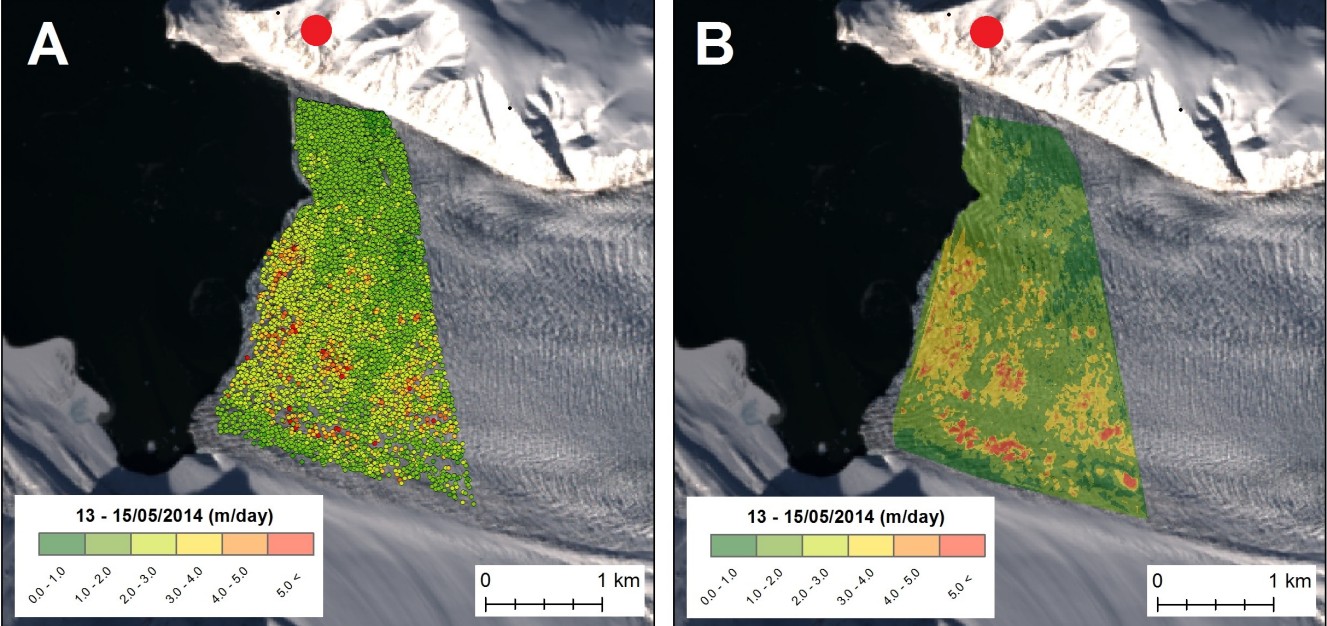

**Figure 6.** An example of PyTrx's feature-tracking and georectification functionality. Points have been tracked between an oblique time-lapse image pair taken between 13 and 15 May 2014 at Kronebreen camera K2. The raw point positions and associated velocities (A) can be interpolated to create velocity maps of a given area (B). The red point denotes the location of the time-lapse camera (Kronebreen camera K2). This example is provided with PyTrx at https://github.com/PennyHow/PyTrx.

1991). Between two images, the position of a pixel ($I$) will change ($\delta x, \delta y$) over time ($\delta t$), assuming that the pixel intensity is unchanging (Zhang and Chanson, 2018):

$$I(x,y,t) = I\big(x + \delta x, y + \delta y, t + \delta t\big) \tag{12}$$

Although it is mainly used to predict the velocity and trajectory of an object in an image, it can also be used for tracking motion
5 between images, such as feature-tracking on glacier surfaces (Solem, 2012).

The Lucas-Kanade algorithm approximates the Optical Flow of sparse points from image to image using a search window that is typically 3×3 pixels (Lucas and Kanade, 1981), assuming that all 9 points in each window have the same motion. It is effective at measuring small displacements because of its thorough examination in a given window. This is ideal when dealing with slow-moving glaciers or high interval frequency sequences (e.g. more than one image every day). However, ice velocities
10 derived using this method (such as shown in Fig. 6) also include signals of crevasse propagation and surface deformation. This is a key reason why dense feature-tracking methods are often preferred for deriving ice velocities over sparse methods; as the template grid ensures that measured displacements are purely associated with ice motion.

The Lucas Kanade Optical Flow approximation algorithm is available in the *calcOpticalFlowPyrLK* function in the OpenCV library, which is used in PyTrx because of its computational efficiency. It is often used in real-time video photogrammetry.





Tracking is implemented between each image pair, rather than continual tracking through the sequence. The first image in each pair is assigned as the reference image where points are seeded, and the second image is the destination image.

As noted in the previous section, photogrammetric measurements can be non-replicable because of the inclusion of falsely tracked points due to a lack of robust point tracking evaluation. Back-tracking verification (Kalal et al., 2010) is implemented in PyTrx to limit false tracking. Back-tracking verification is used to assess point coherency by tracking points back from the destination image to the reference image. This generates two sets of points in the reference image, the initial seeded points and the corresponding back-tracked points. If a back-tracked point is within a given distance to the position of the seeded point then it is deemed accurate and is kept. Points which exceed this distance threshold are discarded. The distance between the seeded point and the back-tracked point is used as a measure of noise. Together with the signal, this can be used to determine the signal-to-noise (SNR) ratio of each point.

## 4.6 Deriving area and line objects

Current photogrammetric software focus on deriving velocities from time-lapse sequences, such as ImGRAFT (Messerli and Grinsted, 2015) and Pointcatcher (James et al., 2016). Other measures of the glacial system would be valuable, such as changes in supraglacial lakes, the expression of a meltwater plume, and terminus position. PyTrx has been developed to offer these additional photogrammetric measurements. It can specifically derive area and line/distance measurements, in addition to velocities, from time-lapse sequences. The *Area* and *Length* class objects contain all of the processing steps to obtain these measurements. Both class objects inherit from the *Velocity* class object.

*Area* (found in Measure.py) contains all the processing steps for deriving area measurements from imagery in both an automated and manual manner. The automated detection of areas is based on changes in pixel intensity, from which points are seeded around a detected extent. *Length* (found in Measure.py) contains all the processing steps for manual line/distance measurements from imagery. This is done using the same technique found in the *Area* class object, from which the textitLength class object inherits from. The points defined in both these methods can be translated to real-world area and line objects using the *georectification* information and functions in the *CamEnv* class object.

Functions in the *fileHandler* file can be used to export the data from the area and line objects. The information can be exported as listed point coordinates and areas/distances using the *writeAreaFile* and *writeLineFile* functions. Additionally, the objects can be exported as .shp files using the *writeSHPFile* function for easy importing into mapping software such as ArcMap and QGIS.

Shapefiles have been constructed from the distinguished surface expression of meltwater plumes and overlaid onto a Landsat scene in the example presented in Fig. 7. Meltwater plumes are the main sources of outflow from a tidewater glacier, and tracking their surface area can be used to infer changes in discharge (e.g., How et al., 2017). The steady recession of the meltwater plume extents shown in Fig.7 is linked to diurnal fluctuations in melt production.



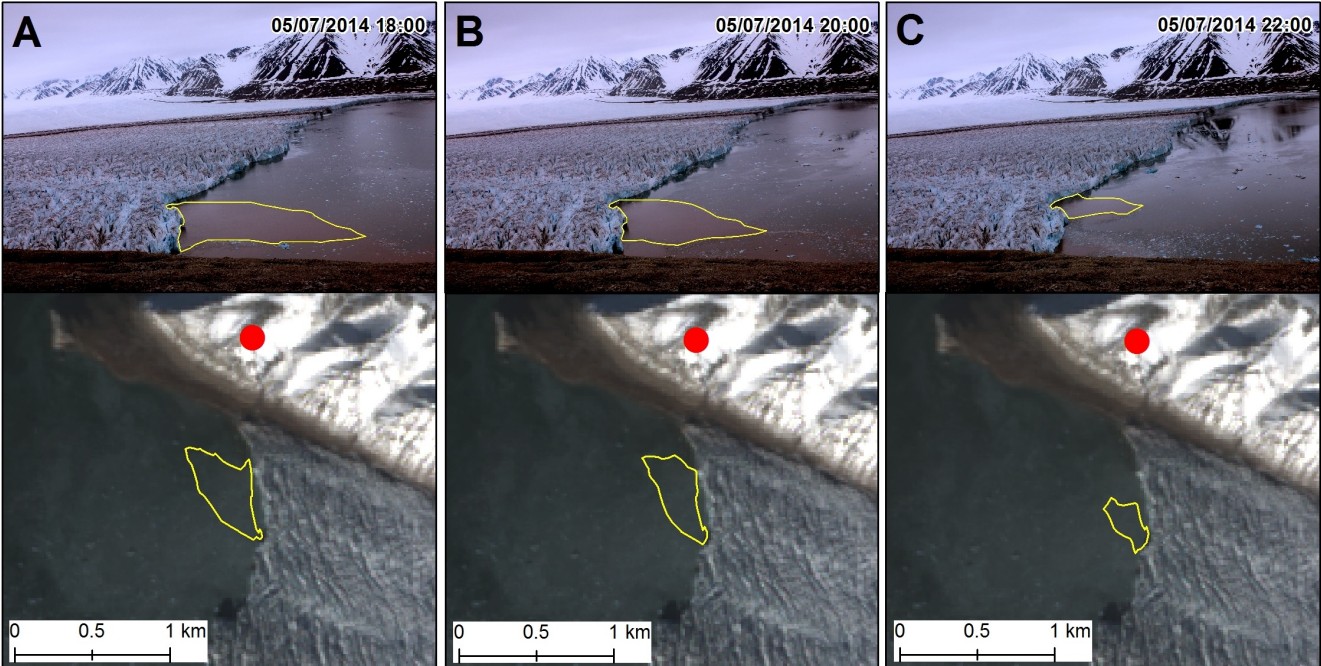

**Figure 7.** Changes in meltwater plume extent distinguished from time-lapse imagery of Kronebreen camera K1. The surface expression of the meltwater plume has been tracked through images captured on 05 July 2014 at 18:00 (A), 20:00 (B), and 22:00 (C) to demonstrate its diurnal recession. Each plot shows the plume definition in the image plane (top) and its translation to real-world coordinates (bottom). A similar example of this is provided with PyTrx at https://github.com/PennyHow/PyTrx.

### 4.6.1  Automated detection

*Area* contains functions for automated detection of areas. Areas are automatically detected based on pixel intensity within the image plane. This entails several key steps and functions to ensure adequate detection in each image. The image is masked to the area of interest firstly, thus reducing processing time and limiting the chance of false detection. The mask can be defined or read from file using the *readMask* function, which is within the FileHandler.py file. This mask function is also used for seeding points in a given region of an image (as part of the feature-tracking and image registration functionality).

The image is next subjected to the simple arithmetic enhancement method outlined in Section 4.3 to better distinguish the target area. The pixel intensities associated with the target area are defined subsequently as a range of the lowest and highest values. This range can either be pre-defined, or manually defined within the program on a point-and-click basis. Pixels within this intensity range are distinguished and grouped using the OpenCV function *cv2.inRange*. The grouped pixels form regions which are transformed into polygons using the OpenCV function *cv2.contour*. Each point within the polygon(s) is defined by coordinates within the image plane.



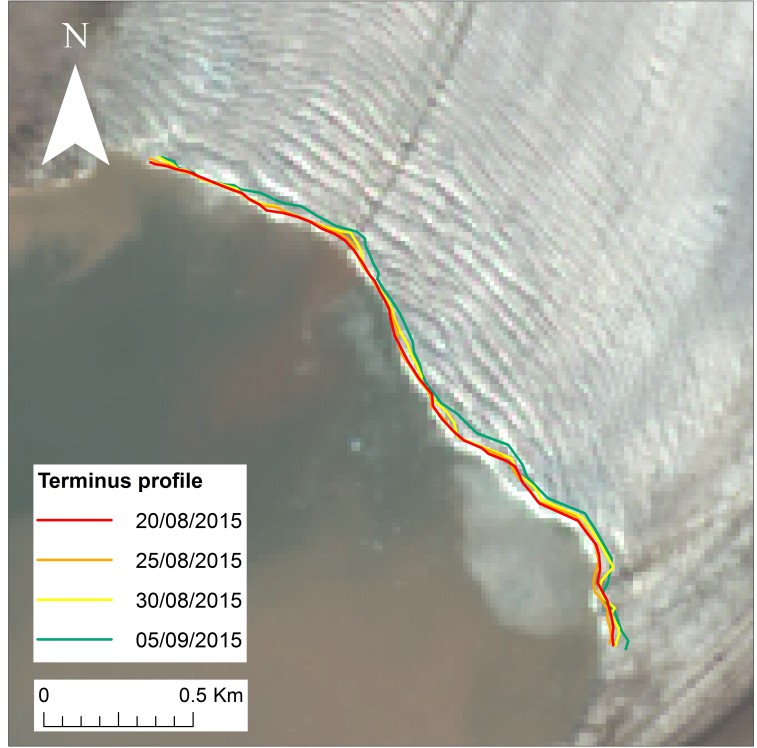

**Figure 8.** An example of PyTrx's ability to the extract sequential terminus profiles from Tunabreen camera T1. A similar example of this is provided with PyTrx at https://github.com/PennyHow/PyTrx.

Often this results in many polygons being created. The number of points in each polygon is used to filter out noise and falsely-detected areas, with small polygons (i.e. constructed with under 40 points) discarded. In addition, the user can define a threshold for the number of polygons retained (i.e. if the threshold is defined as 4, then the 4 largest polygons are retained). There is the additional option to manually verify the detected areas after these steps. This can be defined in the *calcAutoExtents*

5  and *calcAutoAreas* functions with the boolean variable *verify*. This calls on the *verifyExtents* function in PyTrx, which cycles through all the detected extents in all the images and allows the user to manually verify each one based on a click-by-click basis. This can be a time-consuming process with long image sequences, but ensures that falsely-detected areas are discarded.

Using the *calcAutoExtents* function, the detected areas are returned as a set of coordinates for each polygon in the image plane. The *calcAutoAreas* function returns the detected areas as real-world coordinates.

10  ### 4.6.2 Manual detection

Target areas and lines can be defined manually in the coordinate plane of each image in a sequence. This requires the user to click around the area of interest, which creates a set of points from which a polygon/line object is formed. The user input is facilitated by the *ginput* plotting function, which is available in the *matplotlib.pyplot* package. The polygon/line object can





subsequently be georectified to create an object with real-world coordinates. At present, the manual detection functionality allows the user to define one area/line within a given image plane.

An example of PyTrx's manual definition of line features is displayed in Fig. 8. Sequential terminus positions were defined within the image plane on a click-by-click basis, from which line objects were constructed and projected. Terminus profiles have been plotted between 20 August and 05 September 2015 (every five days), providing a detailed record of changes in terminus position over time. This shows a gradual retreat in terminus position over a peak period in the melt season.

Currently, lines are limited to being manually defined, and only one line can be defined within a given image plane. Automated line detection would be a valuable addition in the future for detecting terminus profiles in sequential imagery of calving glacier fronts. However, attempts to detect terminus profiles from oblique time-lapse imagery has proved problematic due to reflections from the adjacent fjord water, changes in tide, and changes in lighting and shadowing. With development, it is hoped that these limitations can be overcome.

## 4.7 Image registration and georectification

*CamEnv* (found in CamEnv.py) compiles and handles all information concerning the camera environment. This includes information concerning the camera calibration, which uses a typical pinhole camera model, and radial and tangential distortion parameters. It also computes the homography from this along with the GCPs and DEM, using a planar projective transformation approach similar to ImGRAFT.

For image registration, the planar projective transformation encapsulates the three-dimensional rotation of the camera platform as movement around its horizontal, vertical and optic axes. The output rotations are used to correct false motion from feature track measurements. Rotation that cannot be accounted for from the two-dimensional displacements is returned as a root-mean-square (RMS) residual pixel value, and this RMS is the main measure of error (i.e. the 'noise'). This is returned along with the velocity (i.e. the 'signal') as a signal-to-noise ratio.

The georectification method follows a similar workflow to ImGRAFT (Messerli and Grinsted, 2015). The homography model is calculated based on a camera model (i.e. extrinsic and intrinsic parameters, and distortion coefficients), for which information can be inputted to PyTrx via a .txt or .mat file format. This information is stored in the *CamEnv* object, and xy points from either the velocity, area or line measurements are subsequently projected onto the DEM in order to obtain their corresponding three-dimensional coordinates.

An example of PyTrx's georectification capabilities, using images from Tunabreen (camera site 1, Fig. 3C), is shown in Fig. 9. Point locations in Fig. 9A denote the position of observed calving events (i.e. the break-off of ice from the glacier terminus) in the image plane. The colour of each point denotes the style of calving, ranging from small break-offs (i.e. waterline and ice fall events) to large collapses (i.e. sheet and stack collapses), and detachments that occur below the waterline (i.e. subaqueous events). These xy point locations have been translated to real-world coordinates using the georectification functions available in PyTrx (Fig. 9B).

The point locations are overlain onto a satellite scene of the glacier terminus, which was captured as close as possible to the time of image capture (Fig. 9B). The point locations tightly follow the terminus position, demonstrating good accuracy in the



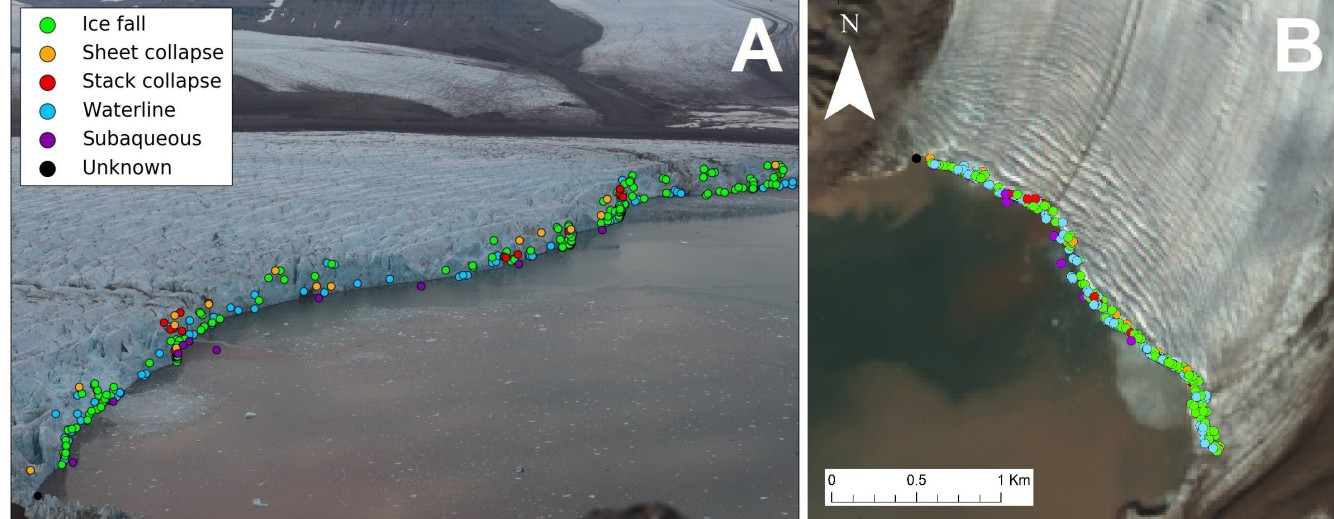

**Figure 9.** Calving events observed in the image plane (A) and georectified (B), with the colour of the point denoting the style of calving. Events were manually detected, from which the style of calving was interpreted. The time-lapse image is taken from a time-lapse sequence captured between the 7th and 8th August 2015. Figure adapted from How et al. (In Review). This example is provided with PyTrx at https://github.com/PennyHow/PyTrx.

georectification technique and the given information about the camera environment. However, points tend to deviate from the terminus position on the eastern side of the terminus, which is the furthest away from the camera. Similar deviation is evident in Fig. 8 also. This may indicate a degree of distance decay that is difficult to correct in the homography model. Distance decay is evident in other georectification methods, especially when performing georectification from monoscopic set-ups (James et

5 al., 2016).

## 5 Evaluation of PyTrx

PyTrx and its modular object-oriented design make it an accessible toolbox for deriving measurements from oblique imagery. PyTrx is flexible and can be adapted easily for the user's requirements, as it is distributed as a set of files with simple example drivers. It broadens the range of photogrammetry toolboxes that are publicly available, with a Python-alternative to those coded

10 in Matlab (Messerli and Grinsted, 2015; James et al., 2016).

The examples shown throughout demonstrate PyTrx's capabilities and range of applications in glaciology. Velocities are derived using an alternative approach, utilising an Optical Flow approximation that proves effective and processing-efficient. The addition of areal and line measurements to PyTrx's outputs have proved valuable in deriving surface areas of supraglacial lakes to show drainage events, and terminus profiles to examine glacial retreat.



The applications presented here also highlight functions to develop in subsequent releases of PyTrx. The detection method for areas proves to be a robust method. However, it was challenging to transfer these to achieve automated line detection from oblique images as alluded to in Section 4.6. This functionality would be useful as an efficient approach to defining terminus positions from oblique time-lapse imagery. It would also be especially valuable for other glaciological applications, such as

detecting grounding line features in satellite imagery (e.g., Christie et al., 2016).

Distance decay in the georectification method is visible in some of the examples shown, specifically where measurements span the entire plane of the image (e.g. figures 8 and 9B). It is possible to limit this with accurate information about the camera environment (e.g. camera location, camera pose) and accurate GCPs that cover the entire image plane. However, this is often not possible as glaciers are challenging environments to conduct photogrammetric measurements. An alternative is to explore

other methods of projective transformations that are more suited to such applications.

Error in PyTrx's feature-tracking approach is largely constrained due to its back-tracking verification algorithm for removing falsely-tracked points. Therefore error estimation is calculated simply as the signal-to-noise ratio; the signal being the tracked displacement, and the noise being the RMS value after image registration. Toolboxes such as ImGRAFT and Pointcatcher use the Monte Carlo method, which is a more robust approach to determining error. The Monte Carlo method uses random repeated

sampling to simulate variation in a system, and has been used in time-lapse photogrammetry to indicate the sensitivity of the image registration to the static point displacements (Messerli and Grinsted, 2015; James et al., 2016). PyTrx could benefit from well-constrained error analysis also. In addition, error analysis for areal and line measurements would be valuable also, which are currently determined using sensitivity analysis (e.g., How et al., 2017).

The glacial photogrammetry toolboxes currently available have aspects and functionality that make them unique and benefi-

cial to use. For instance, ImGRAFT contains sophisticated functions to refine specified components of the camera environment (i.e. camera pose, location, and GCPs) to produce accurate projections (Messerli and Grinsted, 2015). Equally, multiple DEMs can be inputted into Pointcatcher to derive well-constrained vertical and horizontal displacements; even from challenging set-ups where image capture is not perpendicular to ice flow (James et al., 2016). The functions for deriving areal and line measurements (and their programming in Python) are what make PyTrx a unique toolbox. Users therefore have a greater range

of toolboxes to choose from when embarking on glacial photogrammetry.

## 6 Conclusions

The PyTrx toolbox has been presented here to showcase its abilities in obtaining velocity, areal and line measurements from oblique time-lapse imagery. Images were collected using time-lapse cameras installed at Kronebreen and Tunabreen, two tidewater glaciers in Svalbard, from which the functionality of PyTrx could be tested.

The examples shown throughout demonstrate PyTrx's specific applications in deriving ice flow velocities, surface areas of supraglacial lakes and meltwater plumes, georectified point locations denoting the position of calving events, and glacier terminus profiles. PyTrx serves as a Python-alternative to the toolboxes that are currently available, thus widening the applications



of terrestrial photogrammetry in glaciology. Future development of the toolbox has been highlighted, which indicate promise in its growth and applications.

*Code availability.* The PyTrx toolbox is available on GitHub at https://github.com/PennyHow/PyTrx (715 MB). This includes driver scripts for a selection of the examples given in this paper.

*Code and data availability.* Datasets and example driver scripts for the specified examples presented in this paper are available on the PyTrx GitHub repository at https://github.com/PennyHow/PyTrx.

*Author contributions.* PH developed the PyTrx algorithms for obtaining geometric measurements (i.e. areas and lines), and revised the pre-existing functions to produce a coherent, fully documented toolbox. In addition, PH produced all example applications of PyTrx presented in this paper. NRJH developed the homography and georectification functions in PyTrx and supervised LB (née Addison) in developing

the image handling and feature-tracking functionality. LB produced the initial skeleton of PyTrx and its work flow as part of her M.Sc. dissertation in Geographical Information Science, at the University of Edinburgh (2014/15).

*Competing interests.* There are no competing interests present.

*Acknowledgements.* This work is affiliated with the CRIOS project (Calving Rates and Impact On Sea Level), which was supported by the Conoco Phillips-Lundin Northern Area Program. PH is funded by a NERC PhD studentship (reference number 1396698). The DEM

of Kongsfjorden originates from the Norwegian Polar Institute, kindly distributed in a modified form with permission, licensed under the Creative Commons Attribution 4.0 International (CC BY 4.0) license. The DEM of Tempelfjorden originates from ArcticDEM, which can be used and distributed freely. The DEM was created from DigitalGlobe, Inc., imagery and funded under National Science Foundation awards 1043681, 1559691, and 1542736.





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
