# Peer review of "PyTrx: A Python toolbox for deriving velocities, surface areas and line measurements from oblique imagery in glacial environments"

_Geoscientific Instrumentation, Methods and Data Systems, 2018_

## Referee Comment (RC1) · Anonymous Referee #1 · 28 Sep 2018

The authors describe a toolbox for terrestrial photographs directed towards tidewater glacier outlets. It is a combination of personal best practices of the authors, combining different procedures to extract products from these data. Their targeted audiences seem to be students interested in glaciers, without prior knowledge of computer vision nor photogrammetry.

In the wake of open source movements, and the quest for reproducible results the objective of this paper is clear. However, the implementation seems incomplete.

If the intended audiences are students, the implementation has serious limitations. Processing of data can be done with PyTrx, but understanding of the limitations might

not be gained. For example, the velocity estimation is based upon optical flow. This technique (especially the Lucas-Kanade implementation) is highly sensitive to intensity changes. When no movement is present, it can still produce velocities due to over-casting. The weakness in this work is that the authors apply histogram equalization, hereafter optical flow is computed.

If the intended audiences are peers, and the toolbox should be seen as a benchmark to build upon, its structure is limited. In such a case one should expect a modular framework where different methodologies can be interchanged. Now, the processing pipelines of the authors are the only pathway, which might not work for other datasets. For example, the supra glacial lake detection is very simple, while more advanced methods already exist (Koschitzki et al. 2014).

Furthermore, a camera calibration procedure is missing in the toolbox, which makes the toolbox appear incomplete. The paper is similar to (Messerli & Grinstad, 2015), therefor the question arises why the authors do not build upon this effort, and instead a new toolbox is introduced. Furthermore, the presented workflow is based upon methodologies used by the authors for other publications. These methodologies are around for quite some time, and thus the presented work does not advance the field nor does it provide new insights.

Another design issue might puzzle the reader, as the objective of the authors is a toolbox for the glaciological community. However, the implementation is very algorithmic based; the authors implement a sparse point cloud. This will result in a scattered data collection of different locations in space and time. While for modelling a fixed coordinate system would be more sufficient, as in (Ahn & Box, 2010). Also an error budget for the 3D transformation is missing, which in the terrestrial setup this scales with distance, see for example (Schwalbe & Maas 2017).

Lastly, there is a strong tendency towards referencing to Szeliski, which is a book of references, and a Python image processing book of Solem. Off coarse the authors

describe known methodology, but it might have been a bit more specific.

If the former points are implemented it might be a worthwhile contribution. However, this is substantial and asks for a complete restructuring of the toolbox.

minor comments:

p1 l19 "More toolboxes are therefor needed", I disagree with this argument. It is more worthwhile to extent on previous efforts; open codes are available for Imgraft as well as, photogrammetric libraries such as Ames SP and MicMac.

p2 l4 "measurements from photographs" too vague

p2 l5 "photogrammetry" or do you mean signal processing?

p2 l17 "efficient photgrammetry software", to what extent is PyTrx efficient, there is no emphasis placed in the text about it (batch, multithread,...)

p2 l29 maybe change title to put also an emphasis on monoscopic.

p10 l7 "Matlab Computer Vision toolbox", why is camera calibration not included into PyTrx?

p12 l9 Why do the authors not use simple functions, this will increase the versatility of the toolbox.

p14 l2 Why is there manual inspection? Typically, a dataset has a training and a testing set. Hence, why does PyTrx have not the ability to make a "ground truth" and then different methodologies can be tested. This reduces the subjectiveness of manual inspection.

p14 l11 Why not use the HSV space?

p15 l6 the advantage of Shi-Tomasi is its computational efficiency: the determinant does not have to be calculated

p15 l17 Why are sparse point clouds used, and why if (Szeliski) is cited consteantly,

his adaptive region based selection isn't used? Also, I think most products are more helpful if consistent data points are used, then scattered features, seen throughout a scene.

p17 l5 This is by no means new, the authors might have missed to include (Scambos et al. 1992) & (Jeong et al. 2017).

p22 l2 "proves to be robust" loose claim, see testing/training comment above

p22 l11 this backtracking is a relative error. The authors talk about the alternative approach, as implemented by the other toolboxes. These use Monte-Carlo which is an efficient way to grasp propagations of errors, especially in this non-linear system. Thus the authors know of this technique, but implement an inferior method. Why is this done?

p22 l25 "toolboxes to choose from", I don't think it is very efficient as a field to have several implementations. All implementing their own best practice, how do the authors see this as a pro?

Schwalbe & Maas. 2017. The determination of high-resolution spatio-temporal glacier motion fields from time-lapse sequences

Jeong et al. 2017. Improved multiple matching method for observing glacier motion with repeat image feature tracking

Messerli & Grinstad. 2015. Image georectification and feature tracking toolbox: Im-GRAFT

Koschitzki et al. 2014. An autonomous image based approach for detecting glacial lake outburst floods.

Ahn & Box. 2010. Glacier velocities from time-lapse photos: technique development and first results from the Extreme Ice Survey (EIS) in Greenland

Scambos et al. 1992. Application of image cross-correlation to the measurement of

glacier velocity using satellite image data.

---

## Referee Comment (RC2) · Anonymous Referee #2 · 4 Jan 2019

The authors present an image processing toolbox dedicated to glacier studies. Chapter 1 (Introduction) and chapter 2 (background) are clear, but can be merged in a single chapter to avoid some redundancy. Chapter 3 review different aspects to be considered in the post-processing. Anyway, authors refer to many publications issued from their community and should at least cite 2 or 3 papers issued from the image processing community that has work a lot in the past on image calibration and stereo vision (some tools and algorithms mentioned in that paper are issued from this community). What should be discussed in that part is the limitation of existing methods versus the specificity of the applied field studied. At the end of this chapter, authors should add a synthesis of what is addressed and solved in their present work. In chapter 4 : Please

do make scheme of your processing workflow as mentioned in the text. The camera calibration lean on the use method available in a Matlab Toolbox, which method was used. Why not use available resources in open access and integrate them in your tool ? Paragraph 4.1 described the data set used with simplifications. What are the consequences of your interpolation for DEM ? Paragraph 4.3 How the authors integrate the natural illumination in their image processing approach? This part could take also benefit of image segmentation by region approaches (see multi spectral imagery). Paragraph 4.4 requires a good stability of the observation sensor used on site. Paragraph 4.6 Analysis of several sequences coupled with obtained results should confirm the efficiency of the tool. What is the average computing time ? Chapter 5 Could take benefit of ground truth using an outdoor controlled environment coupled with an alternative analysis approach for instance using DinSAR or Stereo vision. Finally, no measurement uncertainty after using this image processing toolbox are mentioned, neither spatial resolution of initial images. Though the authors want to focus on the presentation of their toolbox, it has to be addressed or at least referenced somewhere.

---

## Author Comment (AC1) · 31 Jan 2019

**Response to Anonymous Referee #1**

The authors describe a toolbox for terrestrial photographs directed towards tidewater glacier outlets. It is a combination of personal best practices of the authors, combining different procedures to extract products from these data. Their targeted audiences seem to be students interested in glaciers, without prior knowledge of computer vision nor photogrammetry. In the wake of open source movements, and the quest for reproducible results the objective of this paper is clear. However, the implementation seems incomplete. If the intended audiences are students, the implementation has serious limitations. Processing of data can be done with PyTrx, but understanding of the limitations might not be gained. If the points given are implemented it might be a worthwhile contribution. However, this is substantial and asks for a complete restructuring of the toolbox.

We would like to thank the reviewer for their constructive comments and feedback. From reading their detailed review, it is clear that the reviewer spent a lot of time reading the paper and testing our toolbox, which we are very grateful for. We believe that this feedback has drastically improved the toolbox, with suggestions that have brought crucial improvements to our attention. Amendments to the toolbox have now been released in our GitHub repository as PyTrx v1.1, with corresponding alterations to the manuscript also. These main changes are:

- 1. PyTrx has now been re-structured as recommended by the reviewer. PyTrx's core functionality now exists as independent functions which do not depend on inputs from PyTrx's class objects. By doing so, PyTrx is more flexible and easier to adapt to meet users' needs. An example of using PyTrx's independent functions for deriving velocities has been included as an example driver (driver\_velocity2.py) to demonstrate this implementation;
- 2. The Measure.py script has been split according to the functions and class objects for deriving homography, velocity, area and line measurements. We believe this better differentiates the types of measurements that can be derived using PyTrx;
- 3. Camera calibration functionality has now been added to PyTrx's; functionality. A camera can be calibrated using an inputted set of calibration chessboard images, either using the stand-alone function *calibrateImages* or when initialising the *CamCalib/CamEnv* class object;
- 4. Histogram equalisation is now an optional step, rather than a mandatory step when loading images. This can be toggled using the boolean flag in both the *readImg* function (in FileHandler.py) and the *CamImage* object.

Additionally we have added a section regarding an error budget, as suggested by the reviewer, and endeavoured to find more varied and wide-reaching references.

Details of our response to the reviewer's major and minor comments are outlined subsequently.

**Major comments**

1. The velocity estimation is based upon optical flow. This technique (especially the Lucas-Kanade implementation) is highly sensitive to intensity changes. When no movement is present, it can still produce velocities due to overcasting. The weakness in this work is that the authors apply histogram equalisation, hereafter optical flow is computed.

An Optical Flow Approximation is adopted in PyTrx's feature-tracking functions because it is highly efficient for tracking a large number of points -50,000 points can be tracked between a pair of images in under 10 seconds. This is computationally more efficient than using traditional template matching algorithms, and desperately needed in this current age of big data and batch processing. For instance, the template matching function in Python's OpenCV takes 10 times longer to track 2000 points between a pair of images (on average, based on our own testing whilst developing PyTrx).

The drawback of the Optical Flow Approximation approach, as the reviewer rightfully points out, is that matching is based on pixel intensity change rather than relative change. Matching based on relative pixel change in a given region is what makes template matching a robust and reliable method for feature-tracking. The Optical Flow Approximation is highly sensitive to changes in pixel intensity, such as those caused by changes in lighting and shadowing. This can prove challenging when tracking between time-lapse images in glacial environments, where conditions can change very quickly. However, PyTrx accounts for this and is able to limit false tracking due to these sensitives with effective point filtering based on the magnitude and direction of the displacement. This places a heavy reliance on the selection of the images used for feature-tracking, to make sure that images are consistent in lighting and shadowing. However, image selection is a crucial step in all optical image processing techniques due to changes in illumination and shadowing, and their subsequent impact on relative pixel change and the resulting output (e.g. Messerli and Grinsted; 2015; James et al., 2016; Schwalbe and Maas, 2017).

The reviewer suggests that histogram equalisation of the images may hinder trackability with the Optical Flow Approximation technique. Histogram equalisation is applied to all images in PyTrx to enhance image contrast, by adjusting the image intensities. We do this for two reasons:

- 1. To make corner features in the image more prominent, so more points can be generated/seeded
- 2. To make features more distinguishable from one image to the next, improving their trackability

In all, we found that applying histogram equalisation improves the distinguishing of glacial features (such as supraglacial lakes and terminus lines) and the tracking of glacial features. This is why it is included in PyTrx as a mandatory step. However, we appreciate that users should have the flexibility to choose whether to apply histogram equalisation to their image sequence. For this reason, PyTrx's histogram equalisation is now optional and is defined throughout (i.e. in all functions and class objects) as an input variable.

2. If the intended audiences are peers, and the toolbox should be seen as a benchmark to build upon, its structure is limited. In such a case one should expect a modular framework where different methodologies can be interchanged. Now, the processing pipelines of the authors are the only pathway, which might not work for other datasets. For example, the supra glacial lake detection is very simple, while more advanced methods already exist (Koschitzki et al. 2014).

Our choice to distribute PyTrx as an object-oriented toolbox stems from the current range of publicly available glacial photogrammetry toolboxes and their flexibility. These toolboxes have been either distributed with a rigid graphical user interface (e.g. Pointcatcher, CIAS) that do not allow access to the source code; or have been distributed as raw functions (e.g. ImGRAFT) which requires background knowledge and labour in order to adapt them to a user's needs.

PyTrx has been designed with object-oriented design in order to provide a middle ground, with semi-rigid functionality. PyTrx has rigidity in its design, catering for beginners in coding and those with little time for adapting raw code. However, the reviewer's comment highlights that PyTrx's flexibility is not adequately demonstrated thus far; rightfully pointing out that PyTrx's core functions are reliant on the class objects.

For this reason, we have totally re-structured PyTrx and released this on our GitHub repository as PyTrx v1.1. This new version now has more flexibility, with PyTrx's core functionality detached from its class objects. Velocities, areas, and line features can now be derived from an image pair/single image without any use of PyTrx's class objects (including image enhancement, georectification, exporting, importing, and plotting functions). We have included an additional example driver (named 'driver\_velocity2.py') to demonstrate this, which only uses PyTrx's stand-alone functions to compute glacier surface velocities from a sequence of images. PyTrx's class objects have been adapted for processing measurements from image sequences - effectively, the stand-alone functions are implemented in these class functions and iterated over an entire inputted image sequence. This new flexibility in PyTrx, with the ability to process photogrammetric measurements with both stand-alone functions and class objects, caters for both beginners and advanced computer programmers thereby making PyTrx accessible to all. All of this information has been conveyed in the manuscript with appropriate alterations and added sections, and also updated in the toolbox documentation.

**3. Furthermore, a camera calibration procedure is missing in the toolbox, which makes the toolbox appear incomplete.**

The reviewer highlights that camera calibration functionality would be an incredibly useful addition to the PyTrx toolbox because it is an integral part of monoscopic photogrammetry. Camera calibration is used to define the intrinsic camera matrix which mathematically represents the camera, and to correct images for distortions that are introduced by the camera and the lens. Raw camera calibration algorithms are available openly, such as the Matlab Computer Vision toolbox and OpenCV for Python and C++. However, these algorithms have yet to be incorporated directly into a glacial photogrammetry toolbox that is openly available to users in the glaciology community. By including them here, PyTrx would be the first open-source glacial photogrammetry toolbox to include camera calibration functionality, to our knowledge. For these reasons, we have now included camera calibration functionality in PyTrx v1.1, based on the algorithms provided in the Python version of OpenCV. This is offered as both a stand-alone function and built into the camera environment (CamEnv) class object. In addition, examples of PyTrx's camera calibration functionality are now included with three of the driver scripts provided – the example for detecting surface lakes ('driver\_autoarea.py') and for deriving glacier surface velocities ('driver\_velocity1.py' and 'driver\_velocity2.py').

Calibration is undertaken in PyTrx using a chessboard/checkerboard approach, which is widely used in computer vision and photogrammetry. The corners of a given chessboard in a set of images are used as a grid to define the intrinsic camera matrix and distortions. Examples of these chessboard images are provided in the Examples directory of PyTrx within the 'calib' folder, which correspond to the two example drivers that perform camera calibrations.

The user can provide the file directory to a set of chessboard images as the calibration input, along with the known number of chessboard corners (i.e. rows, columns), which PyTrx subsequently uses to calibrate the camera. This can either be defined in the camera environment text file (.txt) (whereby the CamEnv object recognises the input based on its data structure and proceeds with camera calibration automatically – or inputted directly to the calibrateImages function.

The calibrateImages function returns the intrinsic camera matrix (K), lens disortion coefficients  $(k_1, k_2, p_1, p_2, k_3)$ , and the calibration error estimate, which are used subsequently in PyTrx's image correction and georectification functions. The intrinsic camera matrix consists of the focal length in pixels  $(f_x, f_y)$  and the principal point  $(c_x, c_y)$  as a 3×3 array which is compatible with PyTrx:

$$K = \begin{bmatrix} f_x & 0 & 0\\ s & f_y & 0\\ c_x & c_y & 1 \end{bmatrix}$$
(1)

Skew (s) is not calculated as part of the intrinsic matrix and is assumed to be 0, as adopted by OpenCV's calibrateCamera algorithm – this is common in computer vision given that camera skew is often neglible in modern cameras. The lens distortion parameters are made up of the radial distortion coefficients  $(k_1, k_2, k_3)$  and tangential distortion coefficients  $(p_1, p_2)$ . Three coefficients are used to represent radial distortion in PyTrx. Whilst we realise that up to eight coefficients can be calculated to define radial

distortion, we found that three are more than sufficient and produce the smallest errors for subsequent image correction.

PyTrx's calibrateImages function encapsulates all of the procedures to calibrate a camera, which are primarily taken from the OpenCV toolbox:

- 1. Chessboard corners are detected in each image (using OpenCV's findChessboardCorners algorithm), based on the inputted chessboard corner dimensions
- 2. If all corners of the chessboard are found, the locations of these corners are defined in the image plane to sub-pixel accuracy (using OpenCV's drawChessboardCorners and cornerSubPix algorithms)
- 3. Image plane coordinates for the detected chessboard corners from all imagery are used to calculate the intrinsic camera matrix and lens distortion coefficients (using OpenCV's calibrateCamera algorithm). Firstly, we calibrate a rough camera matrix and distortion coefficients using the raw inputted coordinates. We then optimise these with a second calibration, whereby the principal point that was calculated initially is fixed. By doing this, we refine the camera matrix and distortion coefficients and reduce the errors. From experimenting with this, we found that the principal point is generally the most accurate and reliable output, and therefore we assume that the principal point is correct for the second calibration
- 4. The optimised camera matrix, lens distortion coefficients, and the error estimate associated with the calibration are returned from this function, which are fed back into the camera environment class object

Subsequent to this, users can now also export the calibration outputs using the writeCalibFile function, which can be found in the FileHandler script. The camera matrix and lens distortion coefficients are exported as a text file (.txt), which is compatible with the calibration file import functionality.

This information has now been incorporated into the manuscript, specifically in Section 4.7 (Image registration and georectification) when discussing camera matrices and lens distortion coefficients.

4. The paper is similar to (Messerli & Grinsted, 2015), therefore the question arises why the authors do not build upon this effort, and instead a new toolbox is introduced. Furthermore, the presented workflow is based upon methodologies used by the authors for other publications. These methodologies are around for quite some time, and thus the presented work does not advance the field nor does it provide new insights.

The reviewer highlights that the methods available in PyTrx (i.e. Optical Flow Approximation, georectification, automated and manual feature detection) have been around for quite some time in photogrammetry and computer vision. However, these methods have not been effectively implemented in glaciology, nor have they been made publicly available to the glaciology community. A small range of toolboxes for glacial photogrammetry applications are currently available, yet none are available in Python which is a common coding language used by the glaciology community. PyTrx provides a valid contribution to glaciology in extending the breadth and range of glacial photogrammetry, and making it more accessible to a greater number in the glaciology and wider environmental science community.

ImGRAFT (Messerli & Grinsted, 2015) is a Matlab toolbox that functions as a feature-tracking tool for optical imagery using template matching. ImGRAFT can calculate glacier velocities from both terrestrial and satellite imagery, with additional georectification algorithms for translating velocities from the terrestrial images. It is a very accomplished toolbox for deriving glacier velocities. However, we wanted to perform additional measurements, namely line measurements (e.g. terminus profiles) and area measurements (e.g. meltwater plume extent). Whilst this can be achieved using ImGRAFT, the functions for doing so are not explicitly provided and it would require a lot of coding knowledge and time to write these. These measurements are not the focus of the ImGRAFT toolbox. PyTrx has therefore been developed to meet this need, and we already have active users who are greatly benefiting from its availability in Python.

5. The authors implement a sparse point cloud. This will result in a scattered data collection of different locations in space and time. While for modelling a fixed coordinate system would be more sufficient, as in (Ahn & Box, 2010).

Traditionally, gridded points are used to track glacier features and generate velocity fields using regularly spaced measurements (e.g. Ahn and Box, 2010; Heid and Kääb, 2012; Messerli and Grinsted, 2015; Schwalbe and Maas, 2017). This is suitable for measuring overall glacier movement in a robust and reliable fashion. However, there is a distinct disadvantage to using gridded points for feature-tracking. Corner features prove most effective for tracking from image to image given that they have highly distinguishable pixel intensities. With gridded points, point selection is based on spacing rather than distinguishable corners thereby limiting effective trackability.

As previously stated, feature-tracking is performed in PyTrx using an Optical Flow Approximation method because it is highly efficient for tracking a large number of points – 50,000 points can be tracked between a pair of images in under 10 seconds. With this many points, a glacier surface can be adequately covered through an image sequence, and produce accurate velocity maps; as demonstrated in Figure 6. These velocity maps have a high spatial resolution that are seldom produced using gridded points (and often take much longer to compute with alternative toolboxes). The reviewer's comment regarding inconsistencies with sparse point clouds are not unfounded, but PyTrx implements thorough filtering (including back-tracking verification) and with careful image selection and a thorough inspection of the output (which should be carried out regardless), we believe the benefits of sparse point tracking far outweigh the limitations.

**6. Also an error budget for the 3D transformation is missing, which in the terrestrial setup this scales with distance, see for example (Schwalbe & Maas 2017).**

The reviewer highlights the need for an error budget for the georectification functionality within PyTrx. Reviewer #2 also reiterates this, commenting that measurement uncertainty is needed in order to verify PyTrx's capabilities (final comment, Chapter 5). To limit repetition, we have decided to address these two concerns here and summarise: 1) the sources of error introduced with PyTrx's velocity, area and line measurements; and 2) quantification of each error source.

There are a handful of sources of error that are present when deriving measurements from monoscopic terrestrial photogrammetry. Velocity errors have previously been highlighted by Messerli and Grinsted (2015), James et al. (2016), and Schwalbe & Maas (2017) in the presentation of previous glacial photogrammetry toolboxes. These errors, along with the errors for line and area measurements, are outlined here:

- 1. **Camera motion error**, dictated by the stability of the camera platform and the accuracy of the homography model generated during the image registration process
- 2. **Pixel tracking error** (in the case of measuring glacier velocity), determined primarily by the selection of the image pair, the coherency of trackable corner features between them, and the magnitude of the pixel track relative to the motion in the camera platform (i.e. the signal-to-noise ratio)
- 3. Detection error (in the case of the automated area detection method), determined primarily by

the variability in illumination/shadowing between the images

- 4. Human error (in the case of the manual detection approaches)
- 5. Georectification error, which is inherently linked to
  - (a) The camera model (including focal length and principal point)
  - (b) The corrected image, linked to the accuracy of the lens distortion coefficients
  - (c) The accuracy of the camera's location and pose (i.e. yaw, pitch, roll)
  - (d) The accuracy of the ground control points (GCPs)
  - (e) The accuracy of the DEM
  - (f) The distance between the camera and the feature of interest

The errors associated with 1, 2, 3 and 4 occur during the measurement in the image plane, with 2 relating to velocity measurements, 3 relating to automated detection, and 4 relating to manual identification. These errors represent the pixel error. The error sources associated with georectification (5) occur during the transformation of these measurements into three-dimensional space (Schwalbe and Maas, 2017). These errors represent the three-dimensional error. Whilst these components have inherent errors associated with them, errors can also be accentuated by inaccurate inputs (such as camera location) and challenging image sequences (e.g. with varying illumination and shadowing). For the purpose of this error analysis, we will therefore look at constraining the errors from each of the outlined components using three of the examples presented in the manuscript, namely the velocities derived from Kronebreen (associated with PyTrx's 'driver\_velocity1.py' script), the meltwater plume footprint areas at Kronebreen (associated with PyTrx's 'driver\_manualarea.py' script). Two measures of error will be calculated – the pixel error from the measurements derived in the image plane, and the three-dimensional error associated with the georectification process.

| Error source           | Average velocity | Average area | Average line |
|------------------------|------------------|--------------|--------------|
|                        | error            | error        | error        |
| Camera motion (px)     | 0.5111           | 0.1294       | 0.9863       |
| Pixel tracking $(px)$  | 0.9667           | _            | _            |
| Feature detection (px) | _                | 86.6870      | 18.8170      |
| Total pixel error (px) | 1.4778           | 86.8164      | 19.8033      |
| Total 3D error (%)     | 0.638            | 0.638        | 0.638        |

Camera motion error is calculated as part of PyTrx's *calcHomography* function, representing the mean error magnitude between an image pair. This error is defined as the movement of static feature points in a pair of images that cannot be accounted for by the homography model (i.e. a RMS value). The error values in the table denote the average error across an image sequence.

The pixel tracking error is associated with calculating velocities through PyTrx's velocity functionality. The error is determined using the back-tracking verification approach discussed in the manuscript. The threshold for back-tracking is set by default to 1. pixel, but can be altered in PyTrx's featureTrack function.

Human error is introduced when a user defines areas/line measurements using PyTrx's manual definition functions. Whilst this error is difficult to constrain, we estimate it for our manual definition examples (in the table above) by iterating the manual definition routine over 10 simulations to produce an average variation. This error will vary between measurements (as demonstrated by the two examples in the table), and therefore it is advised to perform this sensitivity test when using the manual definition methods in PyTrx.

The components that determine the error associated with the georectification process are challenging to constrain individually (e.g. Messerli and Grinsted, 2015; Schwalbe and Maas, 2017). We have chosen to evaluate the these errors collectively through ground-truthing, in order to closely examine the propagation of error over space (i.e. with distance from the camera platform, also known as the baseline). Specifically we used satellite imagery taken at the time of the terrestrial image acquisition to compare the positioning of defined features, namely looking at terminus positions relative to coinciding satellite images (such as those presented in Figure 8).

Over all ground-truthing, we found that the average error (i.e. the difference in position between the georectified feature and the same corresponding feature in the satellite image) was  $\pm 0.638\%$ . This error increases with baseline distance, with conservative estimates of 0.2% up to a baseline of 1500 m, and 0.8% for a baseline of 1500–3000 m. For instance, the defined terminus lines depicted in Figure 8 have an average error of  $\pm 2.4$  m over a baseline of 1500–2000 m, but this error grows to an average of  $\pm 7.7$  m beyond a baseline of 2000 m. From 3000 m, the error increases exponentially and is difficult to adequately constrain given that our camera set-ups do not cover further than 3500 m.

Similar to error estimations by Messerli and Grinsted (2015) and Schwalbe and Maas (2017), we would advise to adopt the average georectification error estimate (0.638%) for measurements, and use the baseline-specific error estimates in instances where all measurements are localised to a given baseline distance.

This information has now been added to the text, specifically the section regarding the Evaluation of PyTrx (Section 4).

7. Lastly, there is a strong tendency towards referencing to Szeliski, which is a book of references, and a Python image processing book of Solem. Off coarse the authors describe known methodology, but it might have been a bit more specific.

More specific and varied references have now been added to the manuscript.

**Minor comments**

Page 1, Line 19: 'More toolboxes are therefor needed', I disagree with this argument. It is more worthwhile to extent on previous efforts; open codes are available for Imgraft as well as, photogrammetric libraries such as Ames SP and MicMac.

We suggest in the manuscript that more glacial photogrammetry toolboxes are needed in order to expand the range of toolboxes on offer in different coding languages and with different applications that are beyond calculating glacier velocities. The reviewer rightfully identifies that, in addition to this, it is also worthwhile to focus on expanding pre-established methods.

The reviewer continues by listing examples of 'open code' toolboxes. Whilst we agree that open code is beneficial to users who wish to access and adapt toolboxes, one of the big limitations in monoscopic photogrammetry is the limited range of toolboxes that are open source - i.e. toolboxes that do not require a pricey license in order to operate. For example, ImGRAFT, the photogrammetry toolbox for feature-tracking through satellite scenes and monoscopic set-ups, is programmed in Matlab which requires a license that not all users will have access to.

The reviewer also lists Ames SP and MicMac as other examples of open code toolboxes. Ames SP refers to NASA Ames Stereo Pipeline, which was developed for Multi-View Geometry processing from satellite imagery (Broxton and Edwards, 2008) and has been further applied to generate Digital Elevation Models (DEMs) from stereo satellite imagery (Shean et al., 2016). MicMac is a toolbox for Structure-from-Motion (SfM) processing and has been used for DEM generation from both aerial and terrestrial imagery (Rupnik et al., 2017). These software are open source and require no licensing, but in both these cases their applications are more focused on Multi-View Geometry processing and

Structure-from-Motion rather than monoscopic photogrammetry (i.e. measurements derived from one camera view) in terrestrial settings. As PyTrx is an open-source monoscopic photogrammetry toolbox, multi-camera photogrammetry software (such as Ames SP and MicMac) are incomparable.

For this reason, we have included a passage in the paragraph to express that there should be a greater focus on expanding pre-existing toolboxes, as well as developing new ones: 'In order further glacial photogrammetry techniques, there needs to be greater focus on expanding the capabilities of exisiting toolboxes, and a marked effort to develop new toolboxes which widens the range of data products that can be obtained from time-lapse imagery.' (Page 1, Line 19)

We also stress throughout the manuscript that PyTrx is a monoscopic photogrammetry toolbox (including a change to the title of this manuscript, as suggested by the reviewer) to better convey that this toolbox is not comparable to multi-camera toolboxes and toolboxes that primarily handle satellite imagery.

Page 2, Line 4: 'measurements from photographs' too vague

Wording now changed to make the passage more specific: 'Photogrammetry is defined broadly as the extraction of quantitative measurements from optical imagery...'

Page 2, Line 5: 'photogrammetry' or do you mean signal processing?

Yes, signal processing is part of what is described here, transitioning from earlier traditional techniques to digital signal processing with the introduction of digital cameras and computers with high processing powers. We have now added this to the text.

Page 2, Line 17: 'efficient photgrammetry software', to what extent is PyTrx efficient, there is no emphasis placed in the text about it (batch, multithread,...)

See response to Minor Comment from Reviewer #2 regarding Paragraph 4.6.

Page 2, Line 29: maybe change title to put also an emphasis on monoscopic. See response to minor comment from Page 1, Line 19.

Page 10, Line 7: 'Matlab Computer Vision toolbox', why is camera calibration not included into PyTrx? The Matlab Computer Vision toolbox was initially used to calibrate the cameras because it is a well established toolbox that yields reliable and accurate results. We understand that having camera calibration functionality in PyTrx would be ideal and better encapsulate all photogrammetry processing in one unified toolbox. For this reason, we have developed camera calibration functionality in PyTrx using the Python functions available in OpenCV (see major comments for more details about the camera calibration functions available in PyTrx). This has now been updated in the manuscript.

Page 12, Line 9: Why do the authors not use simple functions, this will increase the versatility of the toolbox.

See section 2 of Major Comments.

Page 14, Line 2: Why is there manual inspection? Typically, a dataset has a training and a testing set. Hence, why does PyTrx have not the ability to make a 'ground truth' and then different methodologies can be tested. This reduces the subjectiveness of manual inspection.

Manual inspection is used in PyTrx to better constrain the automated detection of area features from oblique imagery. Area features encapsulate a wide range of glacier features which have different distinguishing properties, such as supraglacial lakes, surfacing meltwater plumes, and debris features. One of the common approaches to distinguishing these in optical imagery is their pixel intensity, however, we realise that this may not necessarily work all the time given optical images are highly sensitive to changes in illumination and shadowing. For this reason, we offer the manual inspection as an optional verify tool to minimise false detection. The automated area detection method has been used to effectively detect supraglacial lake surface areas (e.g. How et al., 2017), and we offer it with the PyTrx toolbox as a broad and basic detecting tool which users can build upon. The use of training and test datasets are application-specific, and we did not see a benefit to including them in the toolbox as this would detract from the broad tools on offer that are presented in this work. The use of training and test datasets, along with the implementation of Machine Learning algorithms, would be an interesting and exciting development, but it is beyond the scope of what is presented here.

**Page 14, Line 11: Why not use the HSV space?**

PyTrx handles image data in a single band (e.g. grayscale, red, blue, green) in order to uphold computational efficiency and speed. Whilst use of the HSV bands would perhaps be useful to utilise in the detection methods, the main disadvantage is the computational demand in handling three bands of image data. PyTrx's image handling functionality can be altered by the user for this application, but we do not offer it here as it would detract from one of PyTrx's primary benefits, which is its efficiency in batch processing.

Page 15, Line 6: The advantage of Shi-Tomasi is its computational efficiency: the determinant does not have to be calculated

This is helpful information provided by the reviewer which we have now added to the manuscript: 'The Shi-Tomasi Corner Detection method built upon this with a scoring function that does not depend on the calculation of the determinant. Corners are ranked based on the quality level and the minimum Euclidean distance.'

Page 15, Line 17: Why are sparse point clouds used, and why if (Szeliski) is cited consteantly, his adaptive region based selection isn't used? Also, I think most products are more helpful if consistent data points are used, then scattered features, seen throughout a scene.

See Section 5 of Major Comments for reply to the sparse point cloud comment. Reference to Szeliski has been ommited – see Section 7 of Major Comments.

Page 17, Line 5: This is by no means new, the authors might have missed to include (Scambos et al. 1992) & (Jeong et al. 2017).

Agreed. The suggested references have been added to text.

Page 22, Line 2: 'proves to be robust' loose claim, see testing/training comment above. Phrase removed.

Page 22, Line 11: This backtracking is a relative error. The authors talk about the alternative approach, as implemented by the other toolboxes. These use Monte-Carlo which is an efficient way to grasp propagations of errors, especially in this non-linear system. Thus the authors know of this technique, but implement an inferior method. Why is this done?

See response to Major Comment 6.

Page 22, Line 25: 'toolboxes to choose from', I don't think it is very efficient as a field to have several implementations. All implementing their own best practice, how do the authors see this as a pro? See section 4 of Major Comments.

**Response to Anonymous Referee #2**

We would like to thank the reviewer for providing valuable feedback for improving our manuscript. We have provided a thorough response below to each of the reviewer's queries. The main alterations have been the addition of camera calibration functionality to the PyTrx toolbox, the merging of the Introduction and Background sections to avoid repetition, and the inclusion of error estimation (as also requested by Reviewer #1).

**Minor comments**

Chapter 1 (Introduction) and chapter 2 (background) are clear, but can be merged in a single chapter to avoid some redundancy.

Agreed. Chapter 1 and Chapter 2 have been merged together and sections have been merged to reduce repetition.

Chapter 3 review different aspects to be considered in the post-processing. Anyway, authors refer to many publications issued from their community and should at least cite 2 or 3 papers issued from the image processing community that has work a lot in the past on image calibration and stereo vision (some tools and algorithms mentioned in that paper are issued from this community).

We have now amended this section with more specific and varied references, including work focused on image processing and computer vision.

**What should be discussed in that part is the limitation of existing methods versus the specificity of the applied field studied.**

We acknowledge that we are discussing monoscopic photogrammetry methods and toolboxes for analysis of glacial imagery and we have now made this clearer at the beginning of Section 3 by changing its title and opening paragraph.

**At the end of this chapter, authors should add a synthesis of what is addressed and solved in their present work.**

A synthesis has been added to the manuscript, structured as bulleted points to clearly state how PyTrx is unique. However, this has been added to the beginning of Section 4 rather than at the end of Section 3 (as advised) as it was more fitting to add this as an introductory section to the Features and Applications of PyTrx rather than the section on Common Photogrammetric Methods.

"... Specifically, PyTrx has achieved this with the following key features:

- 1. A sparse feature-tracking approach, using Optical Flow approximation and back-tracking verification to efficiently compute accurate and reliable velocities;
- 2. Approaches for deriving areal and line measurements from oblique images, with automated and manual detection methods;
- 3. Camera calibration functionality built in to calculate internal camera matrices and lens distortion coefficients;
- 4. Written in Python, a free and open-source coding language, and provided with simple example applications for easy use;
- 5. Engineered with object-oriented design for efficient handling of large data sets;
- 6. Core methods designed as stand-alone functions which can be used independent of the class objects, making it flexible for users to adapt accordingly.' (Pages 8–9, Lines 28–29 and 1–7)

In chapter 4 : Please do make scheme of your processing workflow as mentioned in the text.

The workflow presented in Figure 4 has been updated to better convey PyTrx's processing workflow, and a reference to it has been added in Section 4.

**The camera calibration lean on the use method available in a Matlab Toolbox, which method was used. Why not use available resources in open access and integrate them in your tool ?**

The reviewer has highlighted to us that an open-source pathway to producing camera calibration parameters is needed to fully uphold PyTrx's open source ethos. For this reason, we have now included a camera calibration method in PyTrx which can be used either as a stand-alone function, or during the initialisation of the CamEnv class object. For more details, please see Major Comment #3 from Reviewer 1 (who made a similar comment)

**Paragraph 4.1 described the data set used with simplifications. What are the consequences of your interpolation for DEM ?**

Interpolation of a DEM mimics a smoother, homogeneous surface. This is common practise in glaciology where the date of the DEM acquisition does not exactly match the date of the image acquisition. By creating a smoother, homogeneous DEM, we eliminate artefacts in the glacier surface that do not reflect the surface at the time of the image acquisition. This information has now been added to this section:

'These DEMs are distributed with PyTrx in a modified form, with each scene clipped to the area of interest, downgraded to 20 m resolution, and smoothed using a linear interpolation method. Interpolation is used here to eliminate artefacts in the glacier surface that do not reflect the surface at the time of image acquisition.' (Page 10, Lines 5–8)

**Paragraph 4.3 How the authors integrate the natural illumination in their image processing approach? This part could take also benefit of image segmentation by region approaches (see multi spectral imagery).**

PyTrx does not contain specific algorithms to correct for changes in natural illumination, like many monoscopic toolboxes for processing glacial imagery (e.g. Pointcatcher, and ImGRAFT) which instead advise selecting imagery at a similar time of day (i.e. similar illumination conditions). Instead, we try to limit false displacements and feature detection (caused by changes in natural illumination and other such factors) by constraining measurements with robust filtering methods such as the back-tracking verification provided in the feature-tracking methods, and the thresholding and verification methods for automated feature detection.

We explored possible approaches for limiting false measurements with multi spectral image analysis, but found that the handling of larger image data detracted from PyTrx's computational efficiency. In all, PyTrx loses its advantage as a toolbox for quick batch processing by implementing this additional analysis. Whilst we agree that this would be a valuable addition, we believe that this is beyond the scope of the work presented here and would be more fitting in a future version of the PyTrx toolbox.

**Paragraph 4.4 requires a good stability of the observation sensor used on site.**

The authors agree that stability in the camera platform is advantageous when deriving measurements between a pair of images taken at different times. However, platform stability is not vital for seeding points – points are only seeded in one image and therefore shifting in the camera platform does not influence the ability to seed points. We have therefore not included this remark in Section 4.4 (Point Seeding), but have decided to include it in Section 4.5 (Deriving velocities from feature-tracking) where velocities and image registration is discussed as it seems more fitting:

These seeded points are tracked between image pairs in PyTrx using the Lucas Kanade Optical Flow Approximation method (Lucas and Kanade, 1981), which works best when there is minimal motion in the camera platform.' (Page 15, Lines 15–17)

Paragraph 4.6 Analysis of several sequences coupled with obtained results should confirm the efficiency

**of the tool. What is the average computing time ?**

Through timing the runtime of PyTrx's example drivers for deriving velocities (using Python's time package), we found that the software can compute the homography (1000 points successfully tracked on average), velocities (30,000 points successfully tracked on average) and georectification of one image pair in 40 seconds, on average. The example driver 'driver\_velocity2.py' computes three image pairs in 300 seconds. This was ran on a Linux computer with 7.5 GB of memory. Whilst it is useful to quantify PyTrx's efficiency, it could be misleading as these times are arbitrary and partly depend on the operating system, a computer's memory, and the number of active core processors. For this reason, we have not added this analysis to the manuscript.

**Chapter 5 Could take benefit of ground truth using an outdoor controlled environment coupled with an alternative analysis approach for instance using DinSAR or Stereo vision.**

The reviewer proposes to add ground-truthing in order to evaluate the accuracy of PyTrx's velocity measurements. Ground-truthing is a viable approach for comparing outputs in order to assess and refine alike measurements. We have implemented ground-truthing to estimate errors from PyTrx's georectification process, as requested by Reviewer #1. The reviewer here suggests ground-truthing PyTrx's velocity results, specifically using an outdoor controlled environment and an alternative processing approach such as DinSAR or stereo vision.

The engineering of an outdoor controlled environment would involve additional field campaigns and primary data collection, shifting the focus of the research presented in this manuscript. Additionally, our studies used monoscopic camera set-ups (i.e. images from one camera) and the use of stereo vision requires a completely different field set-up. Stereo vision is seldom used for deriving glacier velocities because of the difficulties in effectively implementing these field set-ups (e.g. Eiken and Sund, 2012). Therefore stereo vision is not an established approach to conduct ground-truthing, and it is likely that this extra analysis would introduce more uncertainties rather than resolve them. The addition of an outdoor controlled environment and stereo vision in this paper would therefore require a new field campaign to obtain data that is irrelevant to the aim of this paper.

Feature-tracking through optical and SAR satellite imagery is much more common practise for deriving glacier velocities (e.g. Scambos et al., 1992; Heid and Kääb, 2012; Luckman et al., 2015). These methods track large-scale glacier features over large windows/templates, thereby deriving region-based velocities. This region-based tracking approach has also been implemented in glacial photogrammetry toolboxes for deriving velocities from terrestrial imagery (e.g. Messerli and Grinsted, 2015; Schwalbe and Maas, 2017). PyTrx adopts an alternative approach, identifying and tracking individual glacier features on a point-by-point basis. As a result of this, the velocities derived from PyTrx represent small-scale displacements (e.g. surface deformation) as well as region-based movement and is much more fitting for measuring daily and sub-daily displacements. Whilst PyTrx velocities provide highly-detailed displacements that represent localised change, satellite velocities reflect region-based change on a much larger spatial scale. Results from these two approaches are therefore incomparable because they measure different aspects of the glacier system, and ground-truthing with DinSAR velocities would be an unsuitable addition to this research.

Finally, no measurement uncertainty after using this image processing toolbox are mentioned, neither spatial resolution of initial images. Though the authors want to focus on the presentation of their toolbox, it has to be addressed or at least referenced somewhere.

See response to Major Comment 6 from Reviewer #1.

**PyTrx: A Python toolbox for deriving glacier velocities, surface areas and line measurements from oblique monoscopic imageryin glacial environments**

Penelope How1,2\*, Nicholas R. J. Hulton1,2, and Lynne Buie1

1Institute of Geography, School of GeoSciences, University of Edinburgh, Edinburgh, UK 2Department of Arctic Geology, University Centre in Svalbard, Longyearbyen, Norway **Correspondence:** Penelope How (p.how@vork.ac.uk)

**Abstract.** Terrestrial time-lapse photogrammetry is a rapidly growing method for deriving measurements from glacial environments because it provides high spatio-temporal resolution records of change. However, glacial photogrammetry toolboxes are limited currently. Without prior knowledge in photogrammetry and computer coding, they are used primarily to calculate ice flow velocities or to serve as qualitative records. PyTrx (available at https://github.com/PennyHow/PyTrx) is presented

- 5 here as a Python-alternative toolbox to widen the range of monoscopic photogrammetry toolboxes on offer to the glaciology community. The toolbox holds core photogrammetric functions for point seeding, feature-tracking, image registration, and georectification (using a planar projective transformation model). In addition, PyTrx facilitates areal and line measurements, which can be detected from imagery using either an automated or manual approach. Examples of PyTrx's applications are demonstrated using time-lapse imagery from Kronebreen and Tunabreen, two tidewater glaciers in Svalbard. Products from these applications include ice flow velocities surface areas of supraglacial lakes and meltwater plumes, and glacier terminus.
- 10 these applications include ice flow velocities, surface areas of supraglacial lakes and meltwater plumes, and glacier terminus profiles.

**1 Introduction**

Terrestrial time-lapse photogrammetry has proved to be a viable approach for obtaining high spatio-temporal resolution observational records from tidewater glaciers (e.g., Ahn and Box, 2010; Rosenau et al., 2013; James et al., 2014; Pętlicki et al., 2015)

15 photogrammetry is a rapidly growing technique in glaciology as a result of its expanding capabilities, with applications in monitoring change in glacier terminus position (e.g., Kick, 1966), glacier surface conditions (e.g., Parajka et al., 2012; Huss et al., 2013), supraglacial lakes (e.g., Danielson and Sharp, 2013), meltwater plume activity (e.g., Schild et al., 2016; How et al., 2017; Slater et al., 20, and calving dynamics (e.g., Kaufmann and Ladstädter, 2008; Ahn and Box, 2010; James et al., 2014; Whitehead et al., 2014; Petlicki et

20 However, A prevailing application has been in deriving glacier surface velocity from sequential monoscopic imagery using a technique called feature-tracking; as it offers highly detailed (both spatially and temporally) records (e.g., Finsterwalder, 1954; Fox et al., 19

\*Current address: Department of Environment and Geography, University of York, York, UK

, and a handful of software has been developed to perform feature-tracking through terrestrial, monoscopic time-lapse imagery (e.g., Kääb and Vollmer, 2000; Messerli and Grinsted, 2015; James et al., 2016; Schwalbe and Maas., 2017).

Monoscopic time-lapse photogrammetry remains an under-used technique in glaciology because there are few publiclyavailable toolboxes for deriving real-world, meaningful measurements from terrestrial imagery. There is an increasing demand

[revised manuscript text omitted]

**2 Common photogrammetric methods in glaciology**

Current photogrammetry software for monoscopic approaches with glacial imagery can generally be divided into those that perform feature-tracking algorithms such as IMCORR (Scambos et al., 1992), COSI-Corr (Leprince et al., 2007), and CIAS (Kääb and Vollmer, 2000; Heid and Kääb, 2012); and those that perform image translation functions such as Photogeoref (Corripio, 2004) , PRACTISE (Härer et al., 2016), Agisoft PhotoSean, and PhotoModelerand PRACTISE (Härer et al., 2016) . A common limitation is that few pieces of software unite all the photogrammetry processes needed to compute real world measurements from terrestrial monoscopic time-lapse imagery (i.e. distance, area, velocity, and volume and velocity). There are

a handful of toolboxes that provide functions for all of these processes, such as the Computer Vision System toolbox for Matlab and the OpenCV toolbox for C++ and Python. However, these are merely given as stand-alone distributed as raw algorithms and a significant amount of time and knowledge is needed to produce the desired measurements and information.

ImGRAFT (available at )and-imgraft.glaciology.net), Pointcatcher (available at )-lancaster.ac.uk/...pointcatcher.htm) and Environmental Motion Tracking (EMT) (available at at tu-dresden.de/geo/emt/) were the first toolboxes that were made pub-

- 25 licly available and that contain all the processes needed to obtain velocities from terrestrial monoscopic time-lapse imagery set-ups in glacial environments (Messerli and Grinsted, 2015; James et al., 2016). Both are Matlab-based toolboxes using algorithms from the Computer Vision System toolbox, (Messerli and Grinsted, 2015; James et al., 2016; Schwalbe and Maas., 2017). These toolboxes have been developed specifically for glaciological applications, either distributed as raw code (in the case of ImGRAFT) or software with a graphical user interface (in the case of Pointcatcher and EMT). These software follow a similar workflow
- 30 and the steps involved will be discussed subsequently.

**2.1 Image processing**

[revised manuscript text omitted]

$$\quad \left[ \begin{array}{c|c} R & t \end{array} \right] = \left[ \begin{array}{cccc} r_{1,1} & r_{1,2} & r_{1,3} & t_1 \\ r_{2,1} & r_{2,2} & r_{2,3} & t_2 \\ r_{3,1} & r_{3,2} & r_{3,3} & t_3 \end{array} \right]$$
(5)

The intrinsic camera matrix (K) is a 3×3 matrix that contains information about the focal length of the camera in pixels  $(f_x, f_y)$ , the principal point in the image  $(c_x, c_y)$  and the camera skew (s) (Solem, 2012): (Heikkila and Silven, 1997):

$$K = \begin{bmatrix} f_x & 0 & 0 \\ s & f_y & 0 \\ cx & cy & 1 \end{bmatrix}$$
(6)

The given focal length for a fixed focal length lens can be assumed to be precise because there is a limited chance of lens drift.

- 15 It is advised to calculate the focal length for images captured with zoom lenses and compact cameras for greater accuracy. The principal point is also referred to as the optical centre of an image, and describes the intersection of the optical axis and the image plane. Its position is not always the physical centre of the image due to imperfections produced in the camera manufacturing process(?); this difference is known as the principal point offset (Hartley and Zisserman, 2004). The skew coefficient is the measure of the angle between the xy pixel axes, and is a non-zero value if the image axes are not perpendicular
- 20 (i.e. a 'skewed' pixel grid).

The intrinsic camera matrix (K) assumes that the system is a pinhole camera model and does not use a lens to gather and focus light to the camera sensor (Solem, 2012)(Xu and Zhang, 1996). Camera systems that include a lens introduce distortions to the image plane. These distortions are a deviation from a rectilinear projection, in which straight lines in the real world remain straight in an image. These distortions ineffectively represent the target object in the real world, and therefore distortion

25 coefficients  $(k_1, k_2, p_1, p_2, k_3 \dots k_8) \cdot (k_1, k_2, p_1, p_2, k_3 \dots k_5)$  are needed to correct for this. These coefficient values correct for radial  $(k_1, k_2, k_3 \dots k_8) \cdot (k_1, k_2, k_3 \dots k_5)$  and tangential  $(p_1, p_2)$  distortions (Hartley and Zisserman, 2004): (Zhang, 2000):

$$x_{corrected} = x' \left( 1 + k_1 r^2 + k_2 r^4 + k_3 r^6 + k_4 r^8 + k_5 r^{10} \right)$$

$$y_{corrected} = y' \left( 1 + k_1 r^2 + k_2 r^4 + k_3 r^6 + k_4 r^8 + k_5 r^{10} \right)$$
(7)

$$x_{corrected} = x' + \left[2p_1xy + p_2(r^2 + 2x^2)\right]$$

$$y_{corrected} = y' + \left[p_1(r^2 + 2y^2) + 2p_2xy\right]$$
(8)

Where x', y' are the uncorrected pixel locations in an image, and  $x_{corrected}, y_{corrected}$  are their corrected counterparts. Radial distortion arises from the symmetry of the camera lens whilst tangential distortion is caused by misalignment of the camera lens and the camera sensor. Radial distortion is the more apparent type of distortion in images, especially in wide angle images, and those containing straight lines (e.g. skyscraper landscapes) which appear curved. Severe tangential distortion can visibly

alter the depth perception in images.

5

10

The camera matrix and these distortion coefficients can be computed using a set of calibration images taken with the camera that is being used for photogrammetry purposes (Hartley and Zisserman, 2004). These calibration images must be of an object with a known geometry or contain known coordinates (Heikkila and Silven, 1997; Zhang, 2000). A commonly used object is a black and white chessboard, with the positioning and distance between the corners used as the x', y' coordinates (Solem, 2012).

Other objects which can be used are grids of symmetrical circles (with the centre of each circle forming the x', y' coordinates), and targets which are specified by programs that perform camera calibration (e.g. Agisoft Photomodeller).

- The location of the camera and its initial pose (yaw, pitch, roll) are needed to accurately position the camera within the three-dimensional environment. Yaw, pitch and roll are difficult to accurately measure in the field and certain photogrammetry toolboxes calculate and refine yaw, pitch and roll automatically, optimise these parameters, such as ImGRAFT (Messerli and Grinsted, 2015). Camera pose can also be calculated using the principal point (represented as a GCP), along with additional corresponding GCPs to measure the three axes (Fig. 2). The y-axis position of the principal point is used to calculate yaw as an azimuth bearing (Fig. 2A). The *xyz* position of the principal point provides an elevation comparison to the camera location,
- 20 which defines the pitch rotation (Fig. 2B). A GCP along the same x-axis position as the principal point is used to calculate roll (Fig. 2C), signified by the apparent change in elevation (Addison, 2015).

**3 Features and Applications of PyTrx**

PyTrx (available at https://github.com/PennyHow/PyTrx) has been developed to further terrestrial time-lapse photogrammetry techniques in glaciology, and offer an alternative to the monoscopic toolboxes currently available, and address the issues
 outlined previously. Specifically, PyTrx has achieved this with the following key features:

- 1. A sparse feature-tracking approach, using Optical Flow approximation and back-tracking verification to efficiently compute accurate and reliable velocities;
- 2. Approaches for deriving areal and line measurements from oblique images, with automated and manual detection methods;
- 30 3. Camera calibration functionality built in to calculate internal camera matrices and lens distortion coefficients;